# Unified, Practical, and White-box Seismic Tomography with Automatic Differentiation

## Abstract

Seismic tomography methods are complex, diverse, and incompatible with each other. Traditional adjoint approaches are case-specific, requiring challenging analytical derivations for each set of parameters, waves, and loss functions. Approximating wave equation propagation with neural networks (NNs) remains impractical, since finite training datasets cannot cover all seismic parameters for the infinite number of possible geologic models. In this paper, we propose a unified seismic tomography framework with automatic differentiation (AD) for gradient computation, avoiding analytical derivations and NN training. Our framework is designed for generalized misfit functionals and wave equations, supporting broader applications than previous AD-based studies. Our method is fully white-box, and AD gradients are proven to be equivalent to adjoint gradients theoretically and numerically. To show its generality, we performed ten cross-scenario tests across domains (time/frequency), waves (acoustic/SH/P-SV/visco-acoustic/visco-elastic), and losses (waveform/travel time/amplitude). We also evaluated our method on the OpenFWI benchmark dataset to compare with NN methods. Practicality was further demonstrated by a checkerboard test in the Nankai subduction zone, which is challenging for NN methods due to the lack of suitable training datasets. Our method avoids laborious derivation and implementation of adjoint methods, with only modest computational overhead ($1.3$–$1.8\times$ slower and $1.3$–$2.0\times$ more memory without mini-batching or checkpointing in our tests), which can be further reduced with these standard optimizations. We open-sourced a PyTorch-based platform with various extensible wave simulations and imaging methods, facilitating further developments. Our work shows that AD is not merely a tool to avoid manual gradient derivation, as traditionally viewed, but also provides unifying capability, strong practicality, and high interpretability in inverse problems, suggesting broader applications in related fields of scientific computing.

## 1 Introduction

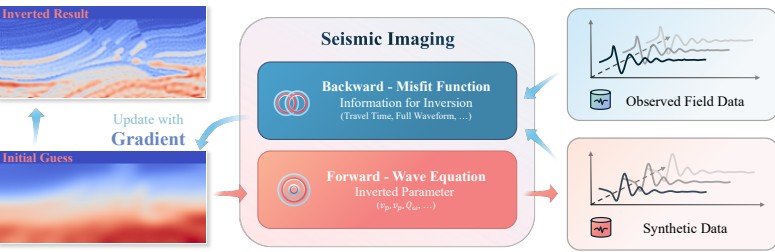

Figure 1: A general gradient-based seismic tomography pipeline. The seismic tomography method and the corresponding gradient are determined by a specific forward and backward combination.

Exploring the subsurface structure is one of humanity's most fundamental pursuits, as it reveals Earth's composition, enables resource exploration, and helps mitigate hazards (Gao, 2011). Moti-

vated by these needs, full-waveform seismic tomography has emerged to transform seismic recordings into detailed subsurface models (Schuster, 2017; Deng et al., 2022). This approach inverts the model by minimizing the seismic data–simulation misfit with the computed gradient, *i.e.*, it seeks the model $\mathbf{m}^* = \arg\min_{\mathbf{m}} J(\mathbf{d}_{\text{obs}}, F(\mathbf{m}))$, where $J(\cdot, \cdot)$ denotes the misfit function measuring the difference between the observed data $\mathbf{d}_{\text{obs}}$ and the simulated data $F(\mathbf{m})$. Although numerous seismic tomography methods exist, each is defined by customizable forward and backward components in Figure 1.

The gradient, quantitatively revealing the model update direction, is at the core of seismic tomography. For computing the gradient, the adjoint method is commonly adopted (Tromp et al., 2005; Liu & Tromp, 2006; Liu, 2020). The analytical gradient and adjoint equation can be derived via the variational principle (see Figure 2) for a given wave equation with selected forward and backward wave propagation modules.

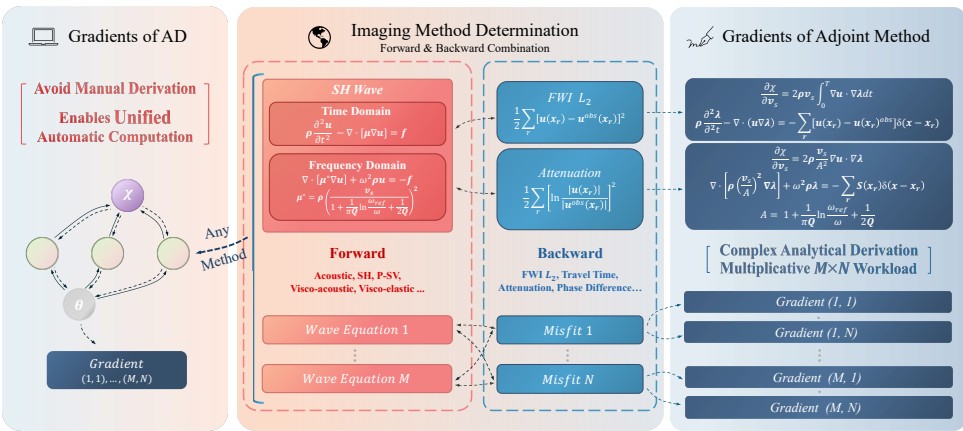

Figure 2: Comparison between traditional adjoint-based imaging and our AD-driven framework. Automatic computational graphs replace manual gradient derivation.

However, despite its efficiency for specialized tomography methods, the adjoint method significantly limits the scope of applications of seismic tomography. Since a comprehensive regional model requires multiple parameters (*e.g.*, seismic P and S wave velocities, and quality factor $Q$), constructing it requires switching among different forward modeling strategies. Moreover, backward modeling methods also vary depending on the task and data quality. Because each analytical gradient derivation from the adjoint method is highly complex and case-specific, frequently changing both forward and backward simulation modules multiplies the derivation workload and significantly increases the overall manual burden (see Figure 2). This limitation wastes useful data and further limits the wider application of seismic tomography (Maurer et al., 2010).

Table 1: Comparison between NN-based methods and our AD-based framework.

|  | NN-based | AD-based (ours) |
| --- | --- | --- |
| **Data Requirements** | Observed data | Observed data |
|  | Prohibitively large dataset for practical use | A rough initial model |
| **Anomaly Recovery** | Over-smoothed results | **High-frequency reconstruction** |
| **Physical Consistency** | Soft loss constraint | Physics-driven |
|  | Not guaranteed | **Always guaranteed** |
| **Interpretability** | Black-box or grey-box models | **Completely white-box** |
| **Practicality** | Limited | **High** |

Neural networks (NNs) seem promising for unifying seismic tomography since they can directly transform seismic recordings into subsurface images without explicit gradients. Recently, many studies have applied various types of NNs to seismic tomography (Zhu et al., 2022; Wu & Lin, 2019; Zhu et al., 2023; Zhang & Lin, 2020; Jin et al., 2021; Feng et al., 2021; Zeng et al., 2021;

Schuster et al., 2024; Desai et al., 2021; Gao et al., 2021; Feng et al., 2024; Feng et al.; Gupta et al., 2024). However, these NNs are trained with specific parameters and structure types (Zhu et al., 2023; Deng et al., 2022), which limits their practical applications. The primary limitation lies in the dataset and model generalizability. Though high-quality, large-scale benchmark datasets exist (Deng et al., 2022; Feng et al., 2023; Li et al., 2024), covering all possible target parameters, wave equations, and possible subsurface structural configurations remains difficult. Constructing such a dataset is like assembling the training data for a universal large language model, but it is particularly challenging for seismic tomography due to the high cost of wave simulations for complex models.

In this work, we propose a unified, practical, and white-box seismic tomography framework based on automatic differentiation (AD). Instead of using NNs to approximate a universal inverse operator, we leverage the underlying gradient computation framework for case-by-case tomography to bypass the dataset limitation. Our framework avoids difficult analytical gradient derivations required by adjoint methods and enables supervised inversion for each case. Our main contributions are:

- Compared to previous AD-based methods for limited misfits in the time domain, we achieve comprehensive unification across time/frequency domains, multiple wave types, and diverse misfit functions.

- We theoretically and numerically demonstrate AD's effectiveness by proving that the gradients from AD are equivalent to those from the analytical adjoint method, regardless of the domain, wave equation, or misfit choices.

- We validate our new framework through experiments across ten diverse scenarios, OpenFWI benchmark experiments, and field checkerboard tests in the Nankai subduction zone.

- We present a comprehensive cost analysis showing that AD avoids laborious derivations and implementations, with only modest overhead within practical limits.

- We provide a customizable seismic tomography platform with various forward and imaging methods, decreasing the practical workload and facilitating new method developments.

## 2 PROBLEM SETUP

Seismic tomography relies on two main forward simulations: time-domain and frequency-domain approaches. Our universal framework considers both methods, and we set up the gradient computing problem separately.

In the time domain, a time-stepping method explicitly discretizes the wave equation. This approach directly simulates wave propagation and is well-suited for capturing time-varying phenomena. The state at time $k$ is computed as

$$\mathbf{h}_k = \mathbf{A}(\boldsymbol{\theta})\mathbf{h}_{k-1} + \mathbf{f}_k, \quad k \geq 1, \tag{1}$$

where $\mathbf{h}_k$ denotes the augmented state vector comprising the current and previous wavefields required. $\mathbf{h}_k$ is compatible with time discretization schemes of arbitrary orders. $\mathbf{h}_0$ is the initial state, $\mathbf{A}(\boldsymbol{\theta})$ is the propagation operator parameterized by the medium properties $\boldsymbol{\theta}$, and $\mathbf{f}_k$ represents the external source.

To avoid confusion, we define the misfit variable $\chi$ and the misfit function $J$ separately. Regardless of the specific misfit function form, the general target gradient is

$$\frac{\partial \chi}{\partial \boldsymbol{\theta}} = \frac{\partial J(\mathbf{h}_1, \ldots, \mathbf{h}_N, \mathbf{d}^{obs})}{\partial \boldsymbol{\theta}}. \tag{2}$$

In the frequency domain, forward modeling is performed by solving the Helmholtz equation at each frequency. It naturally accounts for frequency-dependent information, making it suitable for attenuation imaging (*e.g.*, visco-acoustic wave equation) (Malinowski et al., 2011). Since the solution of the Helmholtz equation represents a steady state, the frequency-domain formulation is inherently stable and does not require any time-domain stability conditions. In addition, it allows for independent frequency computations, which enables efficient parallel processing on GPUs. The forward Helmholtz equation at frequency index $k$ is

$$\mathbf{A}_k(\boldsymbol{\theta})\mathbf{u}_k = \mathbf{s}_k, \tag{3}$$

where $\mathbf{u}_k$ denotes the complex-valued wavefield and $\mathbf{s}_k$ is the corresponding source.

When attenuation is considered, the parameters $\boldsymbol{\theta}$ are complex. By adopting Wirtinger derivatives for complex numbers, the general target gradient expression is

$$\nabla_{\boldsymbol{\theta}}\chi = \begin{bmatrix} \frac{\partial\chi}{\partial\boldsymbol{\theta}_r} \\ \frac{\partial\chi}{\partial\boldsymbol{\theta}_i} \end{bmatrix} = \begin{bmatrix} \frac{\partial\chi}{\partial\boldsymbol{\theta}} + \frac{\partial\chi}{\partial\boldsymbol{\theta}^*} \\ i\left(\frac{\partial\chi}{\partial\boldsymbol{\theta}} - \frac{\partial\chi}{\partial\boldsymbol{\theta}^*}\right) \end{bmatrix}, \tag{4}$$

where $\boldsymbol{\theta}^*$ denotes complex conjugation and the general misfit is defined as

$$\chi = J(\{\mathbf{u}_k(\boldsymbol{\theta},\boldsymbol{\theta}^*), \mathbf{u}_k^*(\boldsymbol{\theta},\boldsymbol{\theta}^*)\}, \mathbf{d}^{obs}). \tag{5}$$

To summarize, our target is to find the gradients in Equation 2 and Equation 4.

## 3 RELATED WORK

Practical applications of NNs are constrained by universal datasets (Section 1). As a more fundamental technique, AD overcomes these limitations by design. AD leverages the chain rule and computational graphs to compute accurate gradients within computer programs (Baydin et al., 2018; Paszke et al., 2017). Instead of training NNs on wide-ranging datasets to approximate a universal inverse operator, AD operates on a case-by-case basis, using predefined physical modules and specific observational data to directly invert for the structure. Moreover, AD offers greater interpretability than NNs since each gradient can be explicitly formulated (Section 5).

Applying AD to seismic tomography is a natural and effective approach for the inherent similarities between the two methodologies (Figure 1). In terms of structure, NNs consist of numerous trainable linear parameters and nonlinear activation functions, whereas seismic tomography focuses on inverting parameters defined on a discrete spatial grid (Zhu et al., 2021). Both approaches begin with a forward pass to compute a misfit (*i.e.*, loss in NNs) and then update the parameters based on the resulting gradient. Physical wave systems can be trained as an analog recurrent neural network (RNN)(Hughes et al., 2019).

Recently, AD has been increasingly adopted in seismic tomography. One research direction leverages AD to simplify specialized imaging methods, primarily for time-domain full-waveform inversion (FWI) (Sambridge et al., 2007; Li et al., 2020; Liu et al., 2024; Cao & Liao, 2015; Zhu et al., 2022; Feng et al., 2023; Wang et al., 2024). In contrast, another line (*e.g.*, ADSeismic (Zhu et al., 2021)) employs AD to develop general seismic tools for tasks such as earthquake location and imaging. However, its tomography application, ADSeismic, is restricted to time-domain FWI $L_2$, addressing only a single type of misfit in one domain. This limitation arises from two key challenges: (1) generalizing AD to handle arbitrary misfit functions (beyond $L_2$ norm) is theoretically difficult, and (2) time-domain and frequency-domain simulations require fundamentally different derivations and implementations. We address both challenges in this paper. Detailed comparison with our method is in Appendix A.

## 4 SEISMIC TOMOGRAPHY VIA AUTOMATIC DIFFERENTIATION

Similar to AD for NNs, we construct a computational graph to compute the gradients (see Figure 3). Each node stores a state variable ($\mathbf{h}_k$, $\mathbf{u}_k$ or $\mathbf{u}_k^*$) and the gradient of the misfit with respect to the node itself. After the forward pass, gradients backpropagate from the misfit through each state variable until reaching the target parameters. Each node computes its local gradient using the chain rule:

$$\frac{\partial\chi}{\partial\mathbf{v}_i} = \sum_{j\in\text{children of }i} \frac{\partial\chi}{\partial\mathbf{v}_j}\frac{\partial\mathbf{v}_j}{\partial\mathbf{v}_i}, \tag{6}$$

where $\mathbf{v}_i$ denotes any state variable. This process ultimately yields $\frac{\partial\chi}{\partial\boldsymbol{\theta}}$ and $\nabla_{\boldsymbol{\theta}}\chi$ in Figure 3. These are precisely the target gradients with respect to seismic parameters in Equation 2 and Equation 4.

## 5 EQUIVALENCE TO THE ADJOINT METHOD

The adjoint method has been proven effective in theory, experiments, and applications (Tromp et al., 2005; Liu & Tromp, 2006; Tape et al., 2009). In this section, we theoretically and numerically demonstrate the equivalence of AD and adjoint gradients to confirm the reliability of our approach.

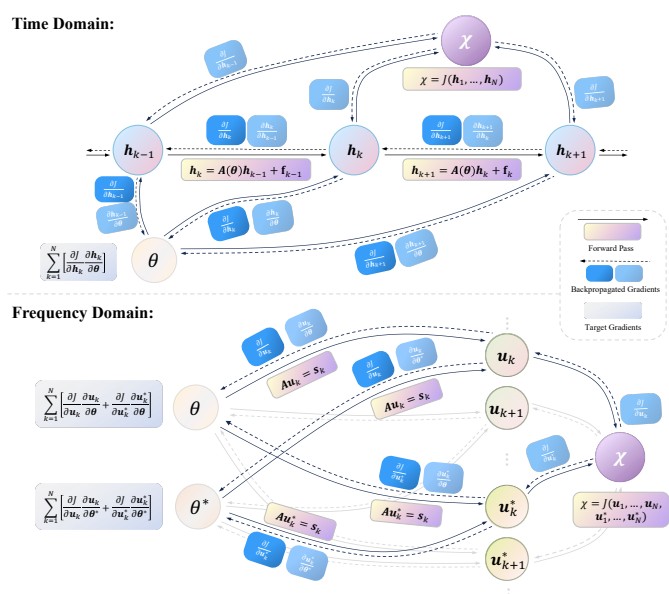

Figure 3: Computational graphs with state nodes. The graph for the time domain is inspired by ADSeismic (Zhu et al., 2021).

## 5.1 THEORETICAL PROOF

**Proposition 1.** *For the general time-domain formulation Equation 1 with misfit in Equation 2, the gradient from the adjoint method equals that from automatic differentiation.*

*Proof.* We now explicitly derive the gradients using both the adjoint method and AD.

**Gradient from the Adjoint Method** By regarding the forward equations as constraints and the misfit function as the objective, the gradient computation can be converted to a nonlinear programming problem (Zhu et al., 2021). Therefore, we introduce the Lagrangian function

$$L = J + \sum_{i=1}^{N} \boldsymbol{\lambda}_i^T \Big( \mathbf{A}\, \mathbf{h}_{i-1} + \mathbf{f}_{i-1} - \mathbf{h}_i \Big), \tag{7}$$

where $\boldsymbol{\lambda}_i^T$ are the Lagrange multipliers or adjoint variables.

Since the forward constraint equations in Equation 1 hold everywhere, adding the derivative of these constraints with respect to $\boldsymbol{\theta}$ to the target gradient in Equation 2 leaves it unchanged. Consequently, the gradient expression can be equivalently written as (details in Appendix B.1):

$$\frac{\partial \chi}{\partial \boldsymbol{\theta}} = \frac{\partial \chi}{\partial \boldsymbol{\theta}} + \sum_{i=1}^{N} \frac{\partial \left( \boldsymbol{\lambda}_i^T \left( \mathbf{A}\mathbf{h}_{i-1} + \mathbf{f}_i - \mathbf{h}_i \right) \right)}{\partial \boldsymbol{\theta}}$$

$$= \sum_{i=1}^{N} \boldsymbol{\lambda}_i^T \frac{\partial \mathbf{A}}{\partial \boldsymbol{\theta}} \mathbf{h}_{i-1} + \sum_{i=1}^{N} \left( \frac{\partial J}{\partial \mathbf{h}_i} - \boldsymbol{\lambda}_i^T + \boldsymbol{\lambda}_{i+1}^T \mathbf{A} \right) \frac{\partial \mathbf{h}_i}{\partial \boldsymbol{\theta}} - \boldsymbol{\lambda}_{N+1}^T \mathbf{A} \frac{\partial \mathbf{h}_N}{\partial \boldsymbol{\theta}}. \tag{8}$$

Differentiating $L$ in Equation 7 with respect to $\mathbf{h}_k$ gives

$$\frac{\partial L}{\partial \mathbf{h}_k} = \frac{\partial J}{\partial \mathbf{h}_k} - \boldsymbol{\lambda}_k^T + \boldsymbol{\lambda}_{k+1}^T \mathbf{A}. \tag{9}$$

Setting the above derivative to zero (the Karush–Kuhn–Tucker conditions) leads to

$$\boldsymbol{\lambda}_k^T = \begin{cases} 0, & k = N+1, \\ \boldsymbol{\lambda}_{k+1}^T \mathbf{A} + \dfrac{\partial J}{\partial \mathbf{h}_k}, & k \leq N. \end{cases} \tag{10}$$

Once the recursive constraints hold, the terms involving $\frac{\partial \mathbf{h}_i}{\partial \boldsymbol{\theta}}$ cancel out in Equation 8, and we obtain

$$\frac{\partial \chi}{\partial \boldsymbol{\theta}} = \sum_{i=1}^{N} \boldsymbol{\lambda}_i^T \frac{\partial \mathbf{A}}{\partial \boldsymbol{\theta}} \mathbf{h}_{i-1}. \tag{11}$$

**Gradient from Automatic Differentiation**    AD in reverse mode relies on the chain rule (see Figure 3). The gradient of the misfit $\chi$ with respect to the model parameters $\boldsymbol{\theta}$ is expressed as

$$\frac{\partial \chi}{\partial \boldsymbol{\theta}} = \sum_{k=1}^{N} \left( \frac{\partial J}{\partial \mathbf{h}_k} \frac{\partial \mathbf{h}_k}{\partial \boldsymbol{\theta}} \right). \tag{12}$$

Given Equation 1, the sensitivity $\frac{\partial \mathbf{h}_k}{\partial \boldsymbol{\theta}}$ is computed recursively as (details in Appendix B.2)

$$\frac{\partial \mathbf{h}_k}{\partial \boldsymbol{\theta}} = \sum_{j=1}^{k} \left( \mathbf{A}^{k-j} \frac{\partial \mathbf{A}}{\partial \boldsymbol{\theta}} \mathbf{h}_{j-1} \right). \tag{13}$$

Substituting the above expression into Equation 12 yields (see Appendix B.3)

$$\frac{\partial \chi}{\partial \boldsymbol{\theta}} = \sum_{k=1}^{N} \frac{\partial J}{\partial \mathbf{h}_k} \left( \sum_{j=1}^{k} \mathbf{A}^{k-j} \frac{\partial \mathbf{A}}{\partial \boldsymbol{\theta}} \mathbf{h}_{j-1} \right) = \sum_{j=1}^{N} \left( \sum_{k=j}^{N} \frac{\partial J}{\partial \mathbf{h}_k} \mathbf{A}^{k-j} \right) \frac{\partial \mathbf{A}}{\partial \boldsymbol{\theta}} \mathbf{h}_{j-1}. \tag{14}$$

Next, we define the adjoint variable as

$$\boldsymbol{\lambda}_j^T \triangleq \sum_{k=j}^{N} \frac{\partial J}{\partial \mathbf{h}_k} \mathbf{A}^{k-j} = \underbrace{\left( \sum_{k=j+1}^{N} \frac{\partial J}{\partial \mathbf{h}_k} \mathbf{A}^{k-(j+1)} \right)}_{\boldsymbol{\lambda}_{j+1}^T} \mathbf{A} + \frac{\partial J}{\partial \mathbf{h}_j}. \tag{15}$$

By recognizing that the underbraced term is precisely $\boldsymbol{\lambda}_{j+1}^T$, we obtain the following recursive relation:

$$\boldsymbol{\lambda}_k^T = \begin{cases} 0, & k = N+1, \\ \boldsymbol{\lambda}_{k+1}^T \mathbf{A} + \frac{\partial J}{\partial \mathbf{h}_k}, & k \leq N. \end{cases} \tag{16}$$

Finally, the overall gradient is given by

$$\frac{\partial \chi}{\partial \boldsymbol{\theta}} = \sum_{j=1}^{N} \boldsymbol{\lambda}_j^T \frac{\partial \mathbf{A}}{\partial \boldsymbol{\theta}} \mathbf{h}_{j-1}. \tag{17}$$

Since Equation 11 and Equation 17 yield the same gradient, and the recursive relations for $\boldsymbol{\lambda}^T$ in Equation 10 and Equation 16 are identical, the two approaches give equivalent gradients.

**Proposition 2.** *For the Helmholtz equation in Equation 3 with a general misfit in Equation 4, the gradient from the adjoint method is identical to that from automatic differentiation.*

*Proof.* For Wirtinger derivatives, the total complex derivatives considering both $\boldsymbol{\theta}$ and $\boldsymbol{\theta}^*$ are:

$$\frac{\partial \chi}{\partial \boldsymbol{\theta}} = \sum_{i=1}^{M} \left( \frac{\partial J}{\partial \mathbf{u}_i} \frac{\partial \mathbf{u}_i}{\partial \boldsymbol{\theta}} + \frac{\partial J}{\partial \mathbf{u}_i^*} \frac{\partial \mathbf{u}_i^*}{\partial \boldsymbol{\theta}} \right), \qquad \frac{\partial \chi}{\partial \boldsymbol{\theta}^*} = \sum_{i=1}^{M} \left( \frac{\partial J}{\partial \mathbf{u}_i} \frac{\partial \mathbf{u}_i}{\partial \boldsymbol{\theta}^*} + \frac{\partial J}{\partial \mathbf{u}_i^*} \frac{\partial \mathbf{u}_i^*}{\partial \boldsymbol{\theta}^*} \right). \tag{18}$$

According to the Product Rule of Wirtinger derivatives, differentiate both sides with respect to $\boldsymbol{\theta}$, which yields:

$$\frac{\partial}{\partial \boldsymbol{\theta}} (\mathbf{A}_i \mathbf{u}_i) = \frac{\partial \mathbf{s}_i}{\partial \boldsymbol{\theta}} \Rightarrow \frac{\partial \mathbf{u}_i}{\partial \boldsymbol{\theta}} = -\mathbf{A}_i^{-1} \frac{\partial \mathbf{A}_i}{\partial \boldsymbol{\theta}} \mathbf{u}_i. \tag{19}$$

Similarly, differentiating with respect to $\boldsymbol{\theta}^*$ and applying the conjugate relationship, we obtain:

$$\frac{\partial \mathbf{u}_i}{\partial \boldsymbol{\theta}^*} = -\mathbf{A}_i^{-1} \frac{\partial \mathbf{A}_i}{\partial \boldsymbol{\theta}^*} \mathbf{u}_i, \qquad \frac{\partial \mathbf{u}_i^*}{\partial \boldsymbol{\theta}} = -(\mathbf{A}_i^*)^{-1} \frac{\partial \mathbf{A}_i^*}{\partial \boldsymbol{\theta}} \mathbf{u}_i^*, \qquad \frac{\partial \mathbf{u}_i^*}{\partial \boldsymbol{\theta}^*} = -(\mathbf{A}_i^*)^{-1} \frac{\partial \mathbf{A}_i^*}{\partial \boldsymbol{\theta}^*} \mathbf{u}_i^*. \tag{20}$$

Substituting the expressions into the derivatives in Equation 18, we get

$$
\begin{aligned}
\frac{\partial \chi}{\partial \boldsymbol{\theta}} &= \sum_{i=1}^{M} \left( \frac{\partial J}{\partial \mathbf{u}_i} \left( -\mathbf{A}_i^{-1} \frac{\partial \mathbf{A}_i}{\partial \boldsymbol{\theta}} \mathbf{u}_i \right) + \frac{\partial J}{\partial \mathbf{u}_i^*} \left( -(\mathbf{A}_i^*)^{-1} \frac{\partial \mathbf{A}_i^*}{\partial \boldsymbol{\theta}} \mathbf{u}_i^* \right) \right), \\
\frac{\partial \chi}{\partial \boldsymbol{\theta}^*} &= \sum_{i=1}^{M} \left( \frac{\partial J}{\partial \mathbf{u}_i} \left( -\mathbf{A}_i^{-1} \frac{\partial \mathbf{A}_i}{\partial \boldsymbol{\theta}^*} \mathbf{u}_i \right) + \frac{\partial J}{\partial \mathbf{u}_i^*} \left( -(\mathbf{A}_i^*)^{-1} \frac{\partial \mathbf{A}_i^*}{\partial \boldsymbol{\theta}^*} \mathbf{u}_i^* \right) \right).
\end{aligned}
\tag{21}
$$

**Gradient from the Adjoint Method**   Similar to the proof in the time domain, we first introduce the Lagrangian function

$$
\mathcal{L} = J(\{\mathbf{u}_i(\boldsymbol{\theta}, \boldsymbol{\theta}^*), \mathbf{u}_i^*(\boldsymbol{\theta}, \boldsymbol{\theta}^*)\}, \mathbf{d}^{obs}) + \sum_{i=1}^{M} \boldsymbol{\lambda}_i^\dagger (\mathbf{A}_i(\boldsymbol{\theta}) \mathbf{u}_i - \mathbf{s}_i) + \sum_{i=1}^{M} \boldsymbol{\Lambda}_i^T (\mathbf{A}_i^*(\boldsymbol{\theta}) \mathbf{u}_i^* - \mathbf{s}_i^*), \tag{22}
$$

where $\boldsymbol{\lambda}_i, \boldsymbol{\Lambda}_i \in \mathbb{C}^N$ are adjoint variables, and $\boldsymbol{\lambda}_i^\dagger$ denotes the conjugate transpose.

Taking Wirtinger derivatives with respect to $\mathbf{u}_i$ gives the adjoint equation:

$$
\frac{\partial \mathcal{L}}{\partial \mathbf{u}_i} = \frac{\partial J}{\partial \mathbf{u}_i} + \boldsymbol{\lambda}_i^\dagger \mathbf{A}_i = 0 \ \Rightarrow\ \frac{\partial J}{\partial \mathbf{u}_i} = -\boldsymbol{\lambda}_i^\dagger \mathbf{A}_i \ \Rightarrow\ \mathbf{A}_i^\dagger \boldsymbol{\lambda}_i = -\left( \frac{\partial J}{\partial \mathbf{u}_i} \right)^\dagger. \tag{23}
$$

Similarly, taking Wirtinger derivatives with respect to $\mathbf{u}_i^*$ yields another adjoint equation:

$$
\frac{\partial J}{\partial \mathbf{u}_i^*} = -\boldsymbol{\Lambda}_i^T \mathbf{A}_i^* \qquad \mathbf{A}_i^T \boldsymbol{\Lambda}_i^* = -\left( \frac{\partial J}{\partial \mathbf{u}_i^*} \right)^\dagger. \tag{24}
$$

Substituting adjoint expressions, the derivatives in Equation 18 are

$$
\frac{\partial \chi}{\partial \boldsymbol{\theta}} = \sum_{i=1}^{M} \left( \boldsymbol{\lambda}_i^\dagger \frac{\partial \mathbf{A}_i}{\partial \boldsymbol{\theta}} \mathbf{u}_i + \boldsymbol{\Lambda}_i^T \frac{\partial \mathbf{A}_i^*}{\partial \boldsymbol{\theta}} \mathbf{u}_i^* \right), \qquad \frac{\partial \chi}{\partial \boldsymbol{\theta}^*} = \sum_{i=1}^{M} \left( \boldsymbol{\lambda}_i^\dagger \frac{\partial \mathbf{A}_i}{\partial \boldsymbol{\theta}^*} \mathbf{u}_i + \boldsymbol{\Lambda}_i^T \frac{\partial \mathbf{A}_i^*}{\partial \boldsymbol{\theta}^*} \mathbf{u}_i^* \right). \tag{25}
$$

**Gradient from Automatic Differentiation**   To compare with adjoint derivatives in Equation 25, we define the adjoint variables $\boldsymbol{\lambda}_i$ and $\boldsymbol{\Lambda}_i$ using the following equations:

$$
\mathbf{A}_i^\dagger \boldsymbol{\lambda}_i = -\left( \frac{\partial J}{\partial \mathbf{u}_i} \right)^\dagger \Rightarrow \frac{\partial J}{\partial \mathbf{u}_i} = -\boldsymbol{\lambda}_i^\dagger \mathbf{A}_i\ , \qquad \mathbf{A}_i^T \boldsymbol{\Lambda}_i^* = -\left( \frac{\partial J}{\partial \mathbf{u}_i^*} \right)^\dagger \Rightarrow \frac{\partial J}{\partial \mathbf{u}_i^*} = -\boldsymbol{\Lambda}_i^T \mathbf{A}_i^*. \tag{26}
$$

By substituting into the derivatives in Equation 21, both the negative signs and the inverse terms cancel pairwise (*e.g.*, $\mathbf{A}_i$ and $\mathbf{A}_i^{-1}$). Thus, the final expressions are given by

$$
\frac{\partial \chi}{\partial \boldsymbol{\theta}} = \sum_{i=1}^{M} \left( \boldsymbol{\lambda}_i^\dagger \frac{\partial \mathbf{A}_i}{\partial \boldsymbol{\theta}} \mathbf{u}_i + \boldsymbol{\Lambda}_i^T \frac{\partial \mathbf{A}_i^*}{\partial \boldsymbol{\theta}} \mathbf{u}_i^* \right), \qquad \frac{\partial \chi}{\partial \boldsymbol{\theta}^*} = \sum_{i=1}^{M} \left( \boldsymbol{\lambda}_i^\dagger \frac{\partial \mathbf{A}_i}{\partial \boldsymbol{\theta}^*} \mathbf{u}_i + \boldsymbol{\Lambda}_i^T \frac{\partial \mathbf{A}_i^*}{\partial \boldsymbol{\theta}^*} \mathbf{u}_i^* \right). \tag{27}
$$

Equation 25 and Equation 27 give the same gradient, and the adjoint equations for $\boldsymbol{\lambda}^T$ and $\boldsymbol{\Lambda}^T$ in Equation 23 and Equation 26 coincide; therefore, the two approaches yield exactly equivalent gradients.

The above proof is valid for arbitrary choices of the wave equation and the misfit function.

## 5.2 NUMERICAL VALIDATION

To show the numerical equivalence on a broad range of scenarios, we conducted experiments on anomaly synthetic models, the Marmousi2 model, and the OpenFWI-B family, with acoustic wave in the time domain and Love wave in the frequency domain.

As shown in Appendix C, across all tested scenarios, these metrics indicate numerical equivalence: correlations and SSIM values are very close to 1 (difference $<10^{-4}$), while Difference Norm and Difference Max consistently remain on the order of $10^{-10}$, which almost reaches floating-point precision. This strong numerical evidence reinforces the equivalence in theory.

## 6 EXPERIMENTS

**Implementation**   We implemented ten tomography scenarios shown in Table 2. Our baseline code is from (1) ADFWI (Liu et al., 2024) for time-domain acoustic and P-SV wave FWI, and (2) a

Table 2: Cross-scenario results. ↑↓: change relative to the initial model.

| Parameter / MS-SSIM↑ Time Domain | FWI $L_2$ | Travel Time |
|---|---|---|
| Acoustic | $v_p$ / 0.982±5.7e-4 (0.115↑) | $v_p$ / 0.887±1.4e-3 (0.019↑) |
| SH | $v_s$ / 0.884±1.3e-3 (0.012↑) | $v_s$ / 0.879±5.8e-4 (0.007↑) |
| P-SV | $v_p$ / 0.896±3.5e-3 (0.029↑) | $v_p$ / 0.877±2.1e-4 (0.010↑) |
| **Frequency Domain** | FWI $L_2$ | Attenuation |
| Visco-acoustic | $Q_p$ / 0.531±2.5e-4 (0.349↑) | $Q_p$ / 0.583±1.2e-3 (0.401↑) |
| Visco-elastic | $Q_s$ / 0.656±8.7e-4 (0.474↑) | $Q_s$ / 0.637±5.1e-4 (0.455↑) |

Matlab-based visco-acoustic wave equation solver (Amini & Javaherian, 2011). Wave equations and misfit expressions used are detailed in Appendix D.

**Cross-scenario Experiments** We validated our unified framework across different scenarios. For time-domain seismic tomography, we employed the classical geometrically complex benchmark Marmousi2 model (Martin et al., 2006). For frequency-domain attenuation imaging, we adopted the Q anomaly model to simulate the Q inversion process following velocity imaging. We introduced MS-SSIM (Multi-Scale Structural Similarity) as the evaluation metric for its consistency in practical applications (Wang et al., 2003; Min et al., 2023) (the advantages over SSIM are discussed in Appendix F). No training set was used, and the experimental settings are provided in Appendix E.3.

Table 2, Figure 4 and Table 23 consistently show successful imaging across different scenarios, demonstrating our method's universality. Gradient visualizations are provided in Appendix H.

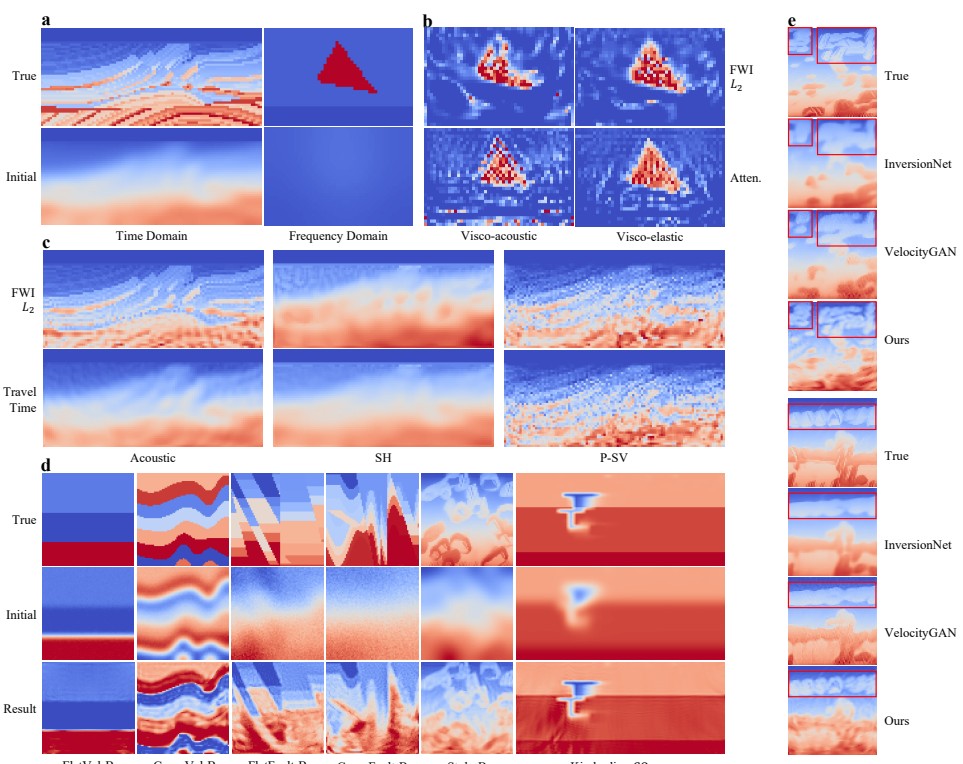

Figure 4: Result illustrations. **a-c**, Cross-scenario experiments (**a**: models, **b**: frequency-domain results, **c**: time-domain results). **d**, OpenFWI experiments. **e**, Detailed comparisons.

**OpenFWI Benchmark Experiments** We compared our method with NN methods: BigFWI (Jin et al., 2024), InversionNet (Wu & Lin, 2019), VelocityGAN (Zhang & Lin, 2020) and UPFWI (Jin et al., 2021) on the OpenFWI Benchmark Dataset (Deng et al., 2022), particularly OpenFWI-B Family (difficult version). The metric of the initial model is set to be worse than the non-outlier results of the NN-based methods for a fair comparison. The experimental settings are detailed in Appendix E.4.

The inversion visualization in Figure 4 demonstrates that our method provides clear imaging of the structures across geological types. Figure 4e further shows that our reconstruction of high-frequency details outperforms that of NN methods.

In the quantitative results (Table 3, 4 and 5), our method scores higher than NN-based methods on models with rich details (*e.g.*, Style-B) but lower on homogeneous models (*e.g.*, FlatFault-B). This inconsistency mainly arises because our physics-driven approach, using a fixed 15 Hz source in OpenFWI, captures complex structures but also introduces extra noise in homogeneous regions. SSIM is the most sensitive to high-frequency artifacts (Appendix F and Appendix G), and it even becomes lower after inversion. Regarding NN-based methods, their results are smoother, which naturally yields higher SSIM values (Figure 4e).

Poor metrics caused by high-frequency noise on homogeneous models do not indicate ineffectiveness in practical applications. First, such artifacts are too minor to affect geological interpretation. As validated in Appendix G, high-frequency artifacts cause the resulting SSIM to be even lower than the initial SSIM, but subsurface structures remain clear and accurately interpretable. Second, perfectly homogeneous regions, as assumed in the synthetic OpenFWI, rarely exist in real scenarios. Cases rich in structural details, such as Style-B, are closer to real geological settings.

Our method does not achieve state-of-the-art performance consistently, but as a unified and practical baseline platform, it shows potential for recovering detailed structures without training sets.

Table 3: OpenFWI benchmark results.

| SSIM↑ | BigFWI-B | BigFWI-M | BigFWI-L | InversionNet | VelocityGAN | UPFWI | **Ours** (init.) |
|---|---|---|---|---|---|---|---|
| FlatVel-B | 0.9658 | 0.9729 | **0.9756** | 0.9356 | 0.9556 | 0.8874 | 0.5673 (0.6978) |
| CurveVel-B | 0.7808 | 0.8053 | **0.8134** | 0.6630 | 0.7111 | 0.6614 | 0.5216 (0.6207) |
| FlatFault-B | 0.8027 | **0.8137** | 0.8033 | 0.7323 | 0.7552 | 0.6937 | 0.6518 (0.6622) |
| CurveFault-B | 0.6781 | **0.6896** | 0.6790 | 0.6137 | 0.6033 | - | 0.5762 (0.5772) |
| Style-B | 0.7567 | 0.7600 | 0.7429 | 0.7667 | 0.7249 | 0.6102 | **0.8093** (0.5552) |
| Kimberlina-$CO_2$ | - | - | - | **0.9872** | 0.9716 | - | 0.9489 (0.7945) |

Table 4: OpenFWI benchmark results.

| MAE↓ | BigFWI-B | BigFWI-M | BigFWI-L | InversionNet | VelocityGAN | UPFWI | **Ours** (init.) |
|---|---|---|---|---|---|---|---|
| FlatVel-B | 0.0233 | 0.0193 | **0.0173** | 0.0304 | 0.0328 | 0.0677 | 0.0395 (0.0402) |
| CurveVel-B | 0.0933 | 0.0816 | **0.0772** | 0.1448 | 0.1428 | 0.1777 | 0.0813 (0.0958) |
| FlatFault-B | 0.0710 | 0.0636 | 0.0644 | 0.0965 | 0.0946 | 0.1416 | **0.0544** (0.0689) |
| CurveFault-B | 0.1245 | 0.1161 | 0.1169 | 0.1705 | 0.1583 | 0.3452 | **0.1034** (0.1313) |
| Style-B | 0.0553 | 0.0538 | 0.0563 | 0.0557 | 0.0649 | 0.1702 | **0.0399** (0.0757) |
| Kimberlina-$CO_2$ | - | - | - | **0.0061** | 0.0119 | - | 0.0103 (0.0193) |

Table 5: OpenFWI benchmark results.

| RMSE↓ | BigFWI-B | BigFWI-M | BigFWI-L | InversionNet | VelocityGAN | UPFWI | **Ours** (init.) |
|---|---|---|---|---|---|---|---|
| FlatVel-B | 0.0696 | 0.0621 | **0.0584** | 0.0680 | 0.0787 | 0.1493 | 0.0718 (0.0879) |
| CurveVel-B | 0.2154 | 0.2006 | **0.1947** | 0.3111 | 0.2611 | 0.3179 | 0.2073 (0.2251) |
| FlatFault-B | 0.1321 | **0.1259** | 0.1269 | 0.1636 | 0.1553 | 0.2220 | 0.1283 (0.1398) |
| CurveFault-B | 0.2027 | 0.1954 | 0.1960 | 0.2507 | 0.2336 | 0.5010 | **0.1918** (0.2050) |
| Style-B | 0.0876 | 0.0867 | 0.0908 | 0.0860 | 0.0979 | 0.2609 | **0.0406** (0.0921) |
| Kimberlina-$CO_2$ | - | - | - | 0.0374 | 0.0387 | - | **0.0195** (0.0403) |

**Field Experiments** To demonstrate our practical application, we tested our method with a Love-wave checkerboard experiment in the Nankai subduction zone (Nakanishi et al., 2008). No training set was used for this experiment. We added checkerboard perturbations to the field model and inverted them from the original field model (Appendix E.5). Figure 5 and Table 18 (SSIM increases from 0.0537 to 0.8812±7.7e-4) show that our method successfully inverted the perturbations and demonstrate potential for field-scale tasks. Such practical tasks are challenging for NN-based methods due to the lack of suitable datasets.

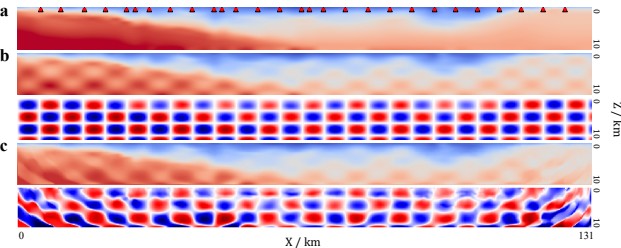

Figure 5: Field checkerboard experiment. **a** Initial / background model. **b** True model with perturbations. **c** Tomography result. Bottom panels of **b** and **c** are differences from the background model.

## 7 COST ANALYSIS

We present a comprehensive cost analysis of the AD method and the adjoint method, summarized in Table 6, covering the entire workflow from analytical derivation to final code execution (details in Appendix I). The results demonstrate that AD avoids laborious derivations and implementations, with only modest computational overhead within practical limits. Employing standard optimization techniques (*e.g.*, mini-batching or checkpointing) can further reduce the overhead.

Table 6: Summary of cost analysis for $m$ wave equations and $n$ misfits.

|  | Adjoint Method | AD (Ours) |
|---|---|---|
| Derivation | $m \times n$ adjoint sources and wavefields | **None** |
| Implementation | $m \times n$ time-reversal solvers and operators | **None** |
| Memory | $1\times$ | 1.3–2.0$\times$ |
| Time | $1\times$ | 1.3–1.8$\times$ |

## 8 CONCLUSION

We present a unified, practical, and white-box seismic tomography framework based on AD, eliminating manual workload while ensuring broad applications to diverse range of misfit functions, wave physics and model parameters. We theoretically and numerically prove that AD-based gradients are equivalent to those from the traditional adjoint method. The generality and practicality of our method are validated across ten diverse scenarios, the OpenFWI dataset and a field checkerboard test in the Nankai subduction zone. Our flexible open-source platform supports direct usage and the development of new methods. Moreover, this work shows that AD is a general and efficient tool for solving scientific inverse problems, which can be extended to more research areas ( *e.g.*, computed tomography (Guzzi et al., 2023; Schoonhoven et al., 2024) and computational fluid dynamics (Zubair et al., 2023)). Future work will focus on: (1) extending our framework to 3D problems; (2) exploring hybrid approaches that leverage NNs for smooth initial model construction, thereby reducing the dependence of physics-based methods on the initial guess (Appendix J). Our method can then be applied to recover the fine structural details that NNs alone cannot capture.

## ETHICS STATEMENT

The authors have read and adhered to the ICLR Code of Ethics. This research contributes to societal well-being by advancing methodologies for natural hazard assessment and fundamental scientific discovery.

To promote responsible stewardship, we offer our framework as a fully transparent, white-box, and open-source platform. This approach ensures reproducibility, encourages verifiable research, and makes advanced scientific tools more accessible.

All experiments were conducted on publicly available benchmark datasets or previously published scientific data, raising no privacy issues. We believe the benefits of this transparent and accessible tool for the scientific community align with the principles of responsible research.

## REPRODUCIBILITY STATEMENT

To ensure reproducibility, we have open-sourced our entire PyTorch-based platform, with the code provided in the supplementary material. The theoretical equivalence between our AD-based method and the traditional adjoint method is proven in Section 5. All experimental settings, including model parameters, source configurations, and computational resources (hardware and software versions), are comprehensively documented in Appendix E and code. The specific wave equations and misfit functions used across our experiments are formally defined in Appendix D.

Furthermore, the supplementary material includes animated visualizations of the forward wave propagation and inversion processes to help in understanding and verification.

## USE OF LLM

Please refer to Appendix 8.

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

## USE OF LLM

In the preparation of this manuscript, we utilized LLMs as a general-purpose assistive tool. The use of LLMs was for the following specific tasks:

- **Language Polishing:** Improving grammar, refining phrasing, and enhancing the overall clarity and readability of the text.

- **Code Assistance:** Debugging code snippets and optimizing parts of the implementation related to our experiments.

## A  COMPARISON WITH ADSEISMIC

Table 7: Comparison of ADSeismic(Zhu et al., 2021) and our method.

| Method | Equivalence Proof | Domain | Wave Types | Misfit | Language |
|---|---|---|---|---|---|
| ADSeismic | Time-domain Forward $+ L_2$ Misfit | Time | Acoustic, P-SV | $L_2$ | Julia |
| Ours | Time- & Frequency-domain Forward + General Misfit | Time & Frequency | General Functionals. (Acoustic, SH, P-SV, Visco-acoustic, Visco-elastic, etc.) | General Functionals. ($L_2$, Travel Time, Attenuation, etc.) | Python (PyTorch) |

## B  DETAILED DERIVATION

### B.1  EQUATION 8

$$
\begin{aligned}
\frac{\partial \chi}{\partial \boldsymbol{\theta}} =& \frac{\partial \chi}{\partial \boldsymbol{\theta}} + \sum_{i=1}^{N} \frac{\partial \left( \boldsymbol{\lambda}_i^T \left( \mathbf{A} \mathbf{h}_{i-1} + \mathbf{f}_i - \mathbf{h}_i \right) \right)}{\partial \boldsymbol{\theta}} \\
=& \sum_{i=1}^{N} \frac{\partial J}{\partial \mathbf{h}_i} \frac{\partial \mathbf{h}_i}{\partial \boldsymbol{\theta}} + \sum_{i=1}^{N} \boldsymbol{\lambda}_i^T \left( \mathbf{A} \frac{\partial \mathbf{h}_{i-1}}{\partial \boldsymbol{\theta}} + \frac{\partial \mathbf{A}}{\partial \boldsymbol{\theta}} \mathbf{h}_{i-1} - \frac{\partial \mathbf{h}_i}{\partial \boldsymbol{\theta}} \right) \\
=& \sum_{i=1}^{N} \boldsymbol{\lambda}_i^T \frac{\partial \mathbf{A}}{\partial \boldsymbol{\theta}} \mathbf{h}_{i-1} + \sum_{i=1}^{N} \left( \frac{\partial J}{\partial \mathbf{h}_i} \frac{\partial \mathbf{h}_i}{\partial \boldsymbol{\theta}} - \boldsymbol{\lambda}_i^T \frac{\partial \mathbf{h}_i}{\partial \boldsymbol{\theta}} \right) + \sum_{i=1}^{N} \left( \boldsymbol{\lambda}_i^T \mathbf{A} \frac{\partial \mathbf{h}_{i-1}}{\partial \boldsymbol{\theta}} \right) \\
=& \sum_{i=1}^{N} \boldsymbol{\lambda}_i^T \frac{\partial \mathbf{A}}{\partial \boldsymbol{\theta}} \mathbf{h}_{i-1} + \sum_{i=1}^{N} \left( \frac{\partial J}{\partial \mathbf{h}_i} \frac{\partial \mathbf{h}_i}{\partial \boldsymbol{\theta}} - \boldsymbol{\lambda}_i^T \frac{\partial \mathbf{h}_i}{\partial \boldsymbol{\theta}} \right) + \sum_{i=1}^{N} \left( \boldsymbol{\lambda}_{i+1}^T \mathbf{A} \frac{\partial \mathbf{h}_i}{\partial \boldsymbol{\theta}} \right) + \boldsymbol{\lambda}_1^T \mathbf{A} \frac{\partial \mathbf{h}_0}{\partial \boldsymbol{\theta}} - \boldsymbol{\lambda}_{N+1}^T \mathbf{A} \frac{\partial \mathbf{h}_N}{\partial \boldsymbol{\theta}} \\
=& \sum_{i=1}^{N} \boldsymbol{\lambda}_i^T \frac{\partial \mathbf{A}}{\partial \boldsymbol{\theta}} \mathbf{h}_{i-1} + \sum_{i=1}^{N} \left( \frac{\partial J}{\partial \mathbf{h}_i} \frac{\partial \mathbf{h}_i}{\partial \boldsymbol{\theta}} - \boldsymbol{\lambda}_i^T \frac{\partial \mathbf{h}_i}{\partial \boldsymbol{\theta}} + \boldsymbol{\lambda}_{i+1}^T \mathbf{A} \frac{\partial \mathbf{h}_i}{\partial \boldsymbol{\theta}} \right) + \boldsymbol{\lambda}_1^T \mathbf{A} \frac{\partial \mathbf{h}_0}{\partial \boldsymbol{\theta}} - \boldsymbol{\lambda}_{N+1}^T \mathbf{A} \frac{\partial \mathbf{h}_N}{\partial \boldsymbol{\theta}} \\
=& \sum_{i=1}^{N} \boldsymbol{\lambda}_i^T \frac{\partial \mathbf{A}}{\partial \boldsymbol{\theta}} \mathbf{h}_{i-1} + \sum_{i=1}^{N} \left( \frac{\partial J}{\partial \mathbf{h}_i} - \boldsymbol{\lambda}_i^T + \boldsymbol{\lambda}_{i+1}^T \mathbf{A} \right) \frac{\partial \mathbf{h}_i}{\partial \boldsymbol{\theta}} - \boldsymbol{\lambda}_{N+1}^T \mathbf{A} \frac{\partial \mathbf{h}_N}{\partial \boldsymbol{\theta}}
\end{aligned}
$$

$$(28)$$

## B.2 EQUATION 13

For $k = 1$, the state update becomes

$$\frac{\partial \mathbf{h}_1}{\partial \boldsymbol{\theta}} = \frac{\partial \mathbf{A}}{\partial \boldsymbol{\theta}} \mathbf{h}_0, \tag{29}$$

where $\mathbf{h}_0$ is the initial state.

For $k = 2$, applying the chain rule to Equation 1 we have

$$\frac{\partial \mathbf{h}_2}{\partial \boldsymbol{\theta}} = \frac{\partial \mathbf{A}}{\partial \boldsymbol{\theta}} \mathbf{h}_1 + \mathbf{A} \frac{\partial \mathbf{h}_1}{\partial \boldsymbol{\theta}}. \tag{30}$$

Substituting Equation 29 into Equation 30 gives

$$\frac{\partial \mathbf{h}_2}{\partial \boldsymbol{\theta}} = \frac{\partial \mathbf{A}}{\partial \boldsymbol{\theta}} \mathbf{h}_1 + \mathbf{A} \frac{\partial \mathbf{A}}{\partial \boldsymbol{\theta}} \mathbf{h}_0. \tag{31}$$

For $k = 3$, we similarly have

$$\frac{\partial \mathbf{h}_3}{\partial \boldsymbol{\theta}} = \frac{\partial \mathbf{A}}{\partial \boldsymbol{\theta}} \mathbf{h}_2 + \mathbf{A} \frac{\partial \mathbf{h}_2}{\partial \boldsymbol{\theta}}. \tag{32}$$

Substituting Equation 31 into Equation 32 yields

$$\frac{\partial \mathbf{h}_3}{\partial \boldsymbol{\theta}} = \frac{\partial \mathbf{A}}{\partial \boldsymbol{\theta}} \mathbf{h}_2 + \mathbf{A} \left( \frac{\partial \mathbf{A}}{\partial \boldsymbol{\theta}} \mathbf{h}_1 + \mathbf{A} \frac{\partial \mathbf{h}_1}{\partial \boldsymbol{\theta}} \right). \tag{33}$$

Recognizing from Equation 29 that $\frac{\partial \mathbf{h}_1}{\partial \boldsymbol{\theta}} = \frac{\partial \mathbf{A}}{\partial \boldsymbol{\theta}} \mathbf{h}_0$, we obtain

$$\frac{\partial \mathbf{h}_3}{\partial \boldsymbol{\theta}} = \mathbf{A}^2 \frac{\partial \mathbf{A}}{\partial \boldsymbol{\theta}} \mathbf{h}_0 + \mathbf{A} \frac{\partial \mathbf{A}}{\partial \boldsymbol{\theta}} \mathbf{h}_1 + \frac{\partial \mathbf{A}}{\partial \boldsymbol{\theta}} \mathbf{h}_2. \tag{34}$$

Extending this recursion to a general time step $k$, we can show that

$$\frac{\partial \mathbf{h}_k}{\partial \boldsymbol{\theta}} = \sum_{j=1}^{k} \left( \mathbf{A}^{k-j} \frac{\partial \mathbf{A}}{\partial \boldsymbol{\theta}} \mathbf{h}_{j-1} \right). \tag{35}$$

## B.3 EQUATION 14

In Equation 14, the summation is taken over the index set

$$S = \{ (k, j) \mid 1 \leq j \leq k \leq N \}. \tag{36}$$

Since addition over a finite set is both commutative and associative, we have

$$\sum_{k=1}^{N} \sum_{j=1}^{k} f(k, j) = \sum_{(k,j) \in S} f(k, j) = \sum_{j=1}^{N} \sum_{k=j}^{N} f(k, j). \tag{37}$$

## C EQUIVALENCE NUMERICAL VALIDATION

To numerically verify the gradient equivalence theoretically established earlier, we compare the misfit gradients computed by our AD framework against those from a traditional adjoint-state method. The comparison was performed across diverse scenarios, including acoustic (time-domain) and Love wave (frequency-domain) simulations on synthetic, Marmousi2, and OpenFWI-B models. For each case, we computed the gradient using both methods and evaluated their similarity via the L2 norm and maximum value of their difference, as well as their correlation and SSIM.

**Time-domain acoustic wave** This section includes results on the anomaly synthetic models (Table 8), the Marmousi2 model (Table 9), and the OpenFWI-B family (Table 10). Gradients are normalized to $[-1, 1]$.

Table 8: Anomaly synthetic models (time-domain acoustic wave).

| Size | Difference Norm | Difference Max | Correlation | SSIM |
|---|---|---|---|---|
| 30×30 | 4.8657e-10 | 4.8061e-10 | 1.00000 | 1.00000 |
| 300×300 | 3.3532e-11 | 2.9799e-11 | 1.00000 | 1.00000 |

Table 9: Marmousi2 model (time-domain acoustic wave).

| Dataset | SSIM |
|---|---|
| Marmousi2 | 0.99996 |

Table 10: OpenFWI-B family (time-domain acoustic wave).

| Dataset | SSIM |
|---|---|
| FlatVel-B | $1.00000 \pm 0.00000$ |
| CurveVel-B | $0.99998 \pm 0.00002$ |
| FlatFault-B | $0.99948 \pm 0.00044$ |
| CurveFault-B | $0.99978 \pm 0.00028$ |
| Style-B | $0.99994 \pm 0.00004$ |
| Kimberlina-CO2 | $0.99998 \pm 0.00006$ |

**Frequency-domain Love wave**  This section includes results on the anomaly synthetic models (Table 11) and the Q anomaly model (Table 12).

Table 11: Anomaly synthetic models (frequency-domain Love wave).

| Size | Difference Norm | Difference Max | Correlation | SSIM |
|---|---|---|---|---|
| 100×100 | 1.329615e-10 | 7.730705e-12 | 1.00000 | 1.00000 |
| 500×500 | 2.523199e-10 | 5.456968e-12 | 1.00000 | 1.00000 |

Table 12: Q anomaly model (frequency-domain Love wave).

| Dataset | SSIM |
|---|---|
| Q anomaly | 1.00000 |

## D  FORWARD AND BACKWARD MODULES

### D.1  WAVE EQUATIONS

#### D.1.1  TIME DOMAIN

**Acoustic Wave**

$$\frac{1}{v_p^2} \frac{\partial^2 p}{\partial t^2} - \nabla^2 p = s, \tag{38}$$

where $p$ is the pressure, $v_p$ is the compressional wave (P wave) speed, and $s$ is the source.

**SH Wave**

$$\rho \frac{\partial^2 u}{\partial t^2} - \nabla \cdot \left[ \mu \nabla u \right] = s, \tag{39}$$

where $u$ denotes the displacement, $\mu$ is the shear modulus, and $\rho$ is the density.

**P-SV Wave** In an isotropic medium, the P-SV system is

$$\frac{\partial \sigma_{xx}}{\partial t} = (\lambda + 2\mu)\,\frac{\partial v_x}{\partial x} + \lambda\,\frac{\partial v_z}{\partial z} + s_{xx},$$

$$\frac{\partial \sigma_{zz}}{\partial t} = \lambda\,\frac{\partial v_x}{\partial x} + (\lambda + 2\mu)\,\frac{\partial v_z}{\partial z} + s_{zz}, \tag{40}$$

$$\frac{\partial \sigma_{xz}}{\partial t} = \mu\left(\frac{\partial v_x}{\partial z} + \frac{\partial v_z}{\partial x}\right) + s_{xz},$$

with velocity update equations

$$\rho\,\frac{\partial v_x}{\partial t} = \frac{\partial \sigma_{xx}}{\partial x} + \frac{\partial \sigma_{xz}}{\partial z} + f_x,$$

$$\rho\,\frac{\partial v_z}{\partial t} = \frac{\partial \sigma_{xz}}{\partial x} + \frac{\partial \sigma_{zz}}{\partial z} + f_z, \tag{41}$$

where

- $\sigma_{xx}$ and $\sigma_{zz}$ are the normal stress components,

- $\sigma_{xz}$ is the shear stress component,

- $v_x$ and $v_z$ denote the particle velocities in the $x$ and $z$ directions,

- $\lambda$ and $\mu$ are the Lamé parameters (with $\mu$ being the shear modulus),

- $\rho$ is the density.

In the time domain, inappropriate inverted parameters can lead to error magnification and divergence. To fundamentally address this issue, all our time-domain simulations strictly adhere to the Courant–Friedrichs–Lewy (CFL) stability condition.

The CFL condition is a necessary condition for the convergence of explicit finite-difference schemes used to solve hyperbolic partial differential equations such as the wave equation. It is typically expressed as

$$dt \leq C \times \frac{dx}{v_{\max}},$$

where:

- $dt$ is the time step,

- $dx$ is the spatial grid spacing,

- $v_{\max}$ is the maximum wave velocity in the model, and

- $C$ is the Courant number (often $\leq 1$, here we select 0.5 as a safe choice).

In our implementation, $dt$ is not chosen arbitrarily but is carefully calculated based on the grid spacing $dx$ and the maximum velocity $v_{\max}$ for each experimental model to ensure the CFL condition is always satisfied. During the inversion process, we continuously check whether the current updated model satisfies the CFL condition. If it does not, gradient clipping is applied to ensure that the CFL condition is maintained as a priority.

### D.1.2 FREQUENCY DOMAIN

**Visco-acoustic Wave** Attenuation and dispersion make the propagation velocity frequency-dependent and complex. In the constant-$Q$ (KF) model, a logarithmic frequency term and an imaginary component are introduced. Thus, the acoustic (P-wave) Helmholtz equation is expressed as

$$\nabla^2 P + \frac{\omega^2}{v_p(\omega)^2}\,P = -S, \tag{42}$$

with the complex velocity defined by

$$\frac{1}{v_p(\omega)} = \frac{1}{v_p} + \frac{1}{\pi \, v_p \, Q} \ln\left(\frac{\omega_{\text{ref}}}{\omega}\right) + \frac{i}{2 \, v_p \, Q}. \tag{43}$$

Here, $v_p$ represents the reference compressional wave speed, $Q$ the quality factor, and $\omega_{\text{ref}}$ a reference frequency.

**Visco-elastic Wave** For viscoelastic media, the displacement is denoted by $U$, and the shear velocity is considered complex and frequency-dependent. The governing SH equation is

$$\nabla^2 U + \frac{\omega^2}{v_s(\omega)^2} \, U = -S, \tag{44}$$

with the KF model defining the complex shear velocity as

$$\frac{1}{v_s(\omega)} = \frac{1}{v_s} + \frac{1}{\pi \, v_s \, Q} \ln\left(\frac{\omega_{\text{ref}}}{\omega}\right) + \frac{i}{2 \, v_s \, Q}. \tag{45}$$

Our frequency-domain forward process is implemented using the Frequency-Domain Finite-Difference (FDFD) method. Specifically, for each frequency $\omega$, a Helmholtz equation of the form

$$A_\omega \, u_\omega = s_\omega$$

is solved. The key steps are as follows:

1. Frequency-Dependent Complex Properties:

   The simulation is based on the Helmholtz equation, which is the frequency-domain representation of the wave equation. To realistically model wave propagation in subsurface media, we incorporate attenuation effects (e.g., the constant-$Q$ Kolsky–Futterman (KF) model), where material properties such as velocity and bulk modulus become complex and frequency-dependent.

   Specifically, in the KF model, the complex velocity is given by

   $$\frac{1}{v(\omega)} = \frac{1}{v} + \frac{1}{\pi v Q} \ln\left(\frac{\omega_{\text{ref}}}{\omega}\right) + \frac{i}{2vQ},$$

   and the complex bulk modulus is

   $$M(\omega) = \rho \cdot [v(\omega)]^2.$$

2. Constructing $A_\omega$:
   - Our method constructs $A_\omega$ as a sparse, complex-valued impedance matrix.
   - Interior: 9-point finite-difference stencil to minimize numerical dispersion.
   - Boundary Condition: Neumann condition for the free surface and PMLs for the other three boundaries, all implemented via complex-stretching coordinates.

3. Solving the Large Sparse Linear System:

   By setting the source term $s_\omega$, the equation $A_\omega u_\omega = s_\omega$ can then be solved. Here, we use PyTorch's built-in solver.

## D.2 MISFIT FUNCTIONS

### D.2.1 TIME DOMAIN

**FWI $L_2$ Misfit** Let $d_{ij}^{\text{obs}}(t)$ and $d_{ij}^{\text{syn}}(t)$ denote the observed and synthetic waveforms for the $i$-th source and $j$-th receiver at time $t$. The waveform $L_2$-norm misfit is defined as

$$\mathcal{J} = \sum_{i=1}^{N} \sum_{j=1}^{M} \sqrt{\sum_{t=1}^{T} \left| d_{ij}^{\text{obs}}(t) - d_{ij}^{\text{syn}}(t) \right|^2}, \tag{46}$$

where $N$ is the number of sources, $M$ the number of receivers per source, and $T$ the number of time steps per trace.

**Travel-time Misfit** Let $d_{ij}^{\text{obs}}(t)$ and $d_{ij}^{\text{syn}}(t)$ denote the observed and synthetic signals for the $i$-th source and $j$-th receiver at time index $t$. Define the cross-correlation function as

$$C_{ij}(k) = \sum_{l=0}^{L-1} d_{ij}^{\text{syn}}(t)\, d_{ij}^{\text{obs}}\Big(t+k\Big), \tag{47}$$

with $k \in \{0, 1, \dots, 2L-2\}$ and $L$ representing the length of the time series for a single trace. The travel-time shift $\tau_{ij}$ is then defined by

$$\tau_{ij} = \operatorname{argmax} C_{ij}(k), \tag{48}$$

and the overall travel-time misfit is given by

$$\mathcal{J} = \frac{1}{2} \sum_{i=1}^{N} \sum_{j=1}^{M} \big|\tau_{ij}\big|^2. \tag{49}$$

### D.2.2 FREQUENCY DOMAIN

**Attenuation Misfit** Let $d_{ij}^{\text{obs}}(k)$ and $d_{ij}^{\text{syn}}(k)$ denote the observed and synthetic complex data in the frequency domain for the $i$-th source and $j$-th receiver at the $k$-th frequency, respectively. The attenuation imaging misfit is defined as

$$\mathcal{J} = \sum_{i=1}^{N} \sum_{j=1}^{M} \sqrt{\sum_{k=1}^{K} \left( \log \frac{|d_{ij}^{\text{obs}}(k)|}{|d_{ij}^{\text{syn}}(k)| + \varepsilon} \right)^2}, \tag{50}$$

where $N$ is the number of sources, $M$ the number of receivers per source, $K$ the number of frequency bins, and $\varepsilon$ is a small constant (*e.g.*, $10^{-10}$) for numerical stability.

## E EXPERIMENTAL SETTINGS

### E.1 COMPUTATIONAL RESOURCES

All the experiments are conducted on a single NVIDIA RTX A5000 GPU with 24 GB. CUDA version is 12.2 and PyTorch version is 2.7.0. The optimizer is Adam.

### E.2 GENERAL SETTINGS

The imaging process is terminated either after a fixed number of iterations or once the misfit reaches a specified threshold. During this process, there is no leakage of the true model. The reported result corresponds to the model obtained at the final iteration, rather than selecting the one with the best performance during optimization.

### E.3 CROSS-SCENARIO EXPERIMENTS

For time-domain seismic velocity imaging, the initial velocity model is obtained by applying a heavy Gaussian blur to the true model. The Gaussian noise parameter is set to 3 for Marmousi2 and 10 for the Q-anomaly model. For frequency-domain attenuation imaging, the initial velocity model is slightly blurred relative to the true model, whereas the initial Q model is heavily blurred.

The free-surface boundary condition is applied to simulate more realistic field conditions. Noise is added to the observed data. Each result is an average of five repetitions under the same settings. Parameters are in Table 13 and Table 14.

Table 13: Time-domain Parameters

| Wave | Size | dx | nt | dt | Source |
|---|---|---|---|---|---|
| Acoustic | 44×100 | 80m | 1500 | 0.006s | 3 Hz |
| SH | 44×100 | 80m | 4000 | 0.004s | 0.5 Hz |
| P-SV | 44×100 | 80m | 1500 | 0.004s | 3 Hz |

Table 14: Frequency-domain Parameters

| Wave | Size | dx | nf | df | Source |
|---|---|---|---|---|---|
| Visco-acoustic | 50×30 | 100m | 24 | 0.25hz | 0.25-6 Hz |
| Visco-elastic | 50×30 | 60m | 24 | 0.25hz | 0.25-6 Hz |

### E.4 OPENFWI BENCHMARK EXPERIMENTS

Table 15: OpenFWI Vel-B, Fault-B and Style-B Family Parameters

| Wave | Size | dx | nt | dt | Source |
|---|---|---|---|---|---|
| Acoustic | 70×70 | 10m | 1000 | 0.001s | 15 Hz |

Table 16: OpenFWI Kimberlina-$CO_2$ Sub-dataset Parameters

| Wave | Size | dx | nt | dt | Source |
|---|---|---|---|---|---|
| Acoustic | 141×401 | 10m | 1250 | 0.002s | 10 Hz |

We compare our method with InversionNet(Wu & Lin, 2019), VelocityGAN(Zhang & Lin, 2020), and UPFWI(Jin et al., 2021).

Following the OpenFWI benchmark experiments, we reproduced the identical acoustic wave settings using a 15 Hz source. Our method is directly applied to 36 models downsampled from the Vel-B, Fault-B and Style-B Family test sets and 15 models from the Kimberlina-$CO_2$ test set without relying on the training dataset.

The initial model is generated by applying a Gaussian blur to the true model. Our misfit function is FWI global correlation. The final statistical results are computed by averaging the performance metrics over the test set.

The performance of deep-learning methods is from OpenFWI (Deng et al., 2022). UPFWI fails on CurveFault-B dataset (SSIM is 0.3941), so we fill it blank. For evaluation, we adopted the SSIM metric following benchmark tests.

Table 17: Field Experiment Parameters

| Wave | Size | dx | nt | dt | Source |
|---|---|---|---|---|---|
| SH | 51×656 | 200m | 8000 | 0.01s | 0.2 Hz |

### E.5 FIELD EXPERIMENT

Using the empirical relationship from (Brocher, 2005), we converted the Vp model described in (Nakanishi et al., 2008) into a Vs model as a field background model.

In the ambient noise tomography field experiment, the ocean bottom stations simultaneously act as virtual sources. We used MS-SSIM (Multi-Scale Structural Similarity) as our evaluation metric (Wang et al., 2003) following Cross-scenario Experiments.

Noise is added to the observed data. Our misfit function is FWI global correlation. Parameters are in Table 15.

Table 18: Field experiment results. ↑↓: change relative to the initial model.

| SSIM ↑ | Initial | Result |
|---|---|---|
| Perturbation | 0.0537 | $0.8812_{\pm 7.7\text{e-}4}$ (0.8275↑) |
| Model | 0.8211 | $0.9609_{\pm 5.9\text{e-}4}$ (0.1398↑) |

## F    SSIM'S VULNERABILITY TO HIGH-FREQUENCY ARTIFACTS

In practical seismic tomography tasks, inverting structures to find anomalies is the central purpose. A metric is needed to evaluate this performance.

SSIM is sensitive to high-frequency artifacts, although such sensitivity does not impact anomaly detection in practical applications.

To demonstrate SSIM's limitation for imaging anomalies under noisy conditions, we generated a series of reconstructions with increasing levels of blur and noise applied to a model containing a known anomaly.

In Table 7, although the reconstructed anomaly appears highly noisy, it can still be easily identified for geological interpretation. However, Table 6 shows that SSIM decreases rapidly even when the anomaly remains clear and interpretable.

We introduce Multi-Scale Structural Similarity (MS-SSIM) (Wang et al., 2003) as a practical metric for seismic tomography tasks. Evaluating image fidelity at multiple resolutions improves the perceptual quality of images (Min et al., 2023). In our experiments, MS-SSIM reflects the recovery of geological anomalies, even when blurred or noisy. This aligns with the real-world goal in seismic exploration: robust detection of subsurface features, rather than producing artificially smooth images that lack detail. Table 6 shows that MS-SSIM is a more robust metric under each noise level, which means MS-SSIM better reflects anomaly detection performance in practical seismic tomography with noise.

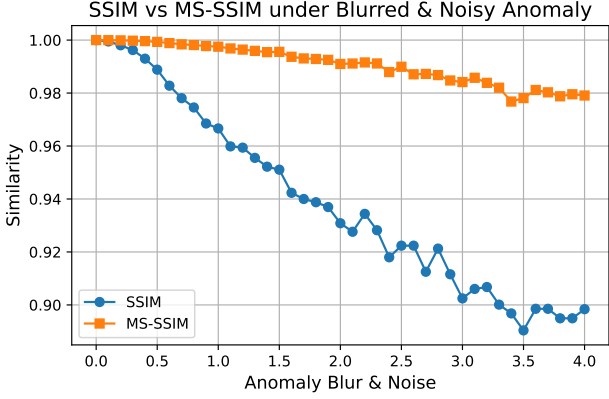

Figure 6: SSIM and MS-SSIM comparison in anomaly detection under noisy conditions.

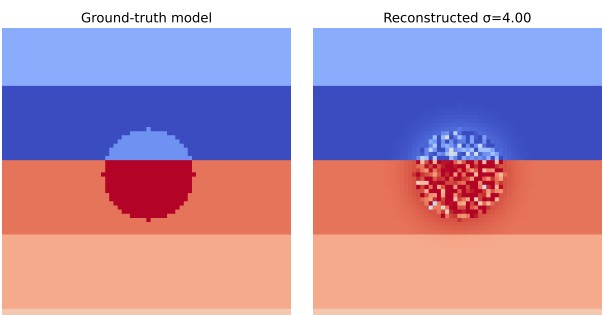

Figure 7: Illustrations of noisy imaging of anomaly.

We further evaluated SSIM and MS-SSIM on reconstructions contaminated with high-frequency noise that does not affect the visual clarity of the geological anomaly. As shown in Figure 9, the model remains clearly visible despite the added noise.

However, the single-scale SSIM score drops sharply in Figure 8. In contrast, MS-SSIM stays essentially constant, demonstrating its robustness to irrelevant noise and its alignment with the true preservation of subsurface features.

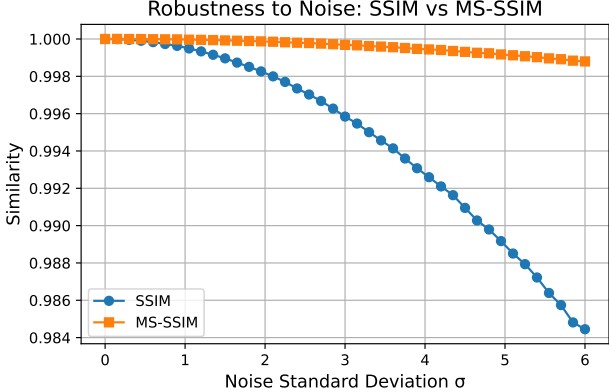

Figure 8: SSIM and MS-SSIM comparison under noisy conditions.

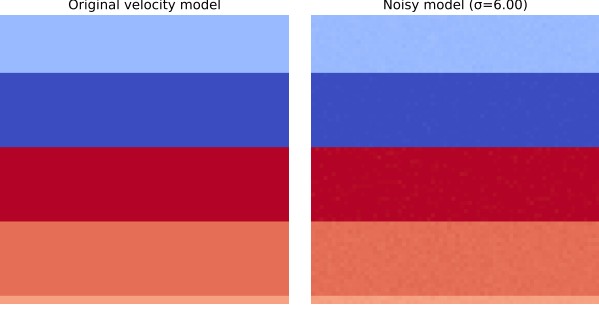

Figure 9: Illustrations of noisy imaging.

To compare the robustness of various metrics (SSIM, MS-SSIM, MAE, and RMSE) to high-frequency noise, we conducted additional experiments. We introduced both high-frequency anomalies and high-frequency noise into the models. If a metric selects the model with gentle blur as the best (where noise is present but anomalies are preserved), it demonstrates robustness to

high-frequency noise while maintaining the ability to detect the desired anomalies. Conversely, if a metric selects the model with strong blur (where both noise and anomalies are removed) as the best, it indicates that the metric is affected by high-frequency noise.

We constructed a series of models, illustrated in Figure 10. Smaller blur corresponds to more high-frequency anomalies and noise, while larger blur corresponds to fewer.

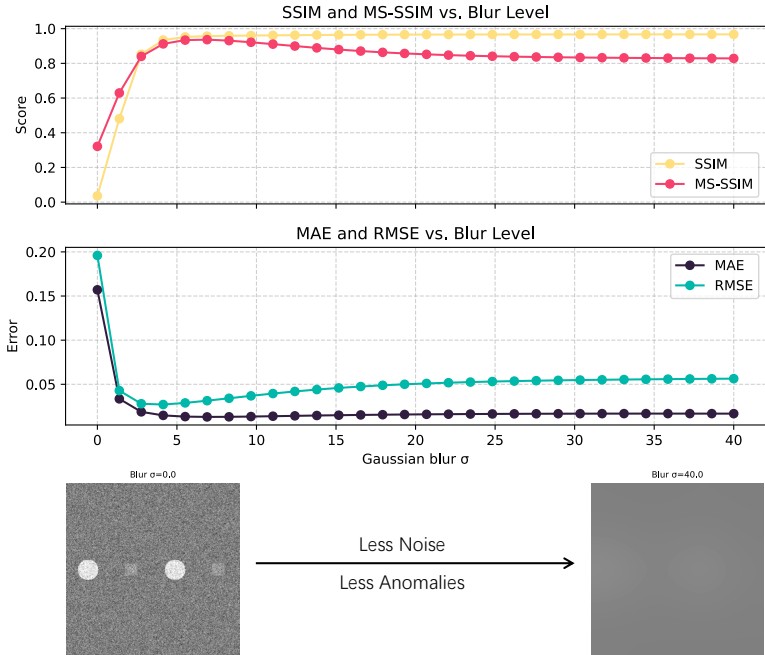



Figure 10: Results of metric robustness to high-frequency noise.



Figure 11: Best model selected by each metric.

The experimental results (Figure 10 and Figure 11) indicate that SSIM tends to select the model with the strongest blur — the one in which both high-frequency noise and high-frequency anomalies are removed. That is, when a method can detect anomalies but also introduces some high-frequency noise (such as our method), SSIM tends to produce a low value, thereby misjudging the method. This is highly non-robust and impractical, as seismic imaging without anomaly details fails. This behavior is non-robust and impractical, as seismic imaging without detailed information fails.

In contrast, the other metrics (MS-SSIM, MAE, and RMSE) choose models that retain high-frequency anomalies, demonstrating greater robustness to high-frequency noise.

# G    INCONSISTENCY BETWEEN LOW SSIM AND RECOVERED STRUCTURES

In the OpenFWI benchmark experiment, although detailed features can be clearly recovered, the SSIM drops significantly due to high-frequency noise in the uniform regions. High-frequency artifacts affect our SSIM metric.

For example, the black boxes highlight the layer boundaries in Figure 12, but the SSIM largely decreases.

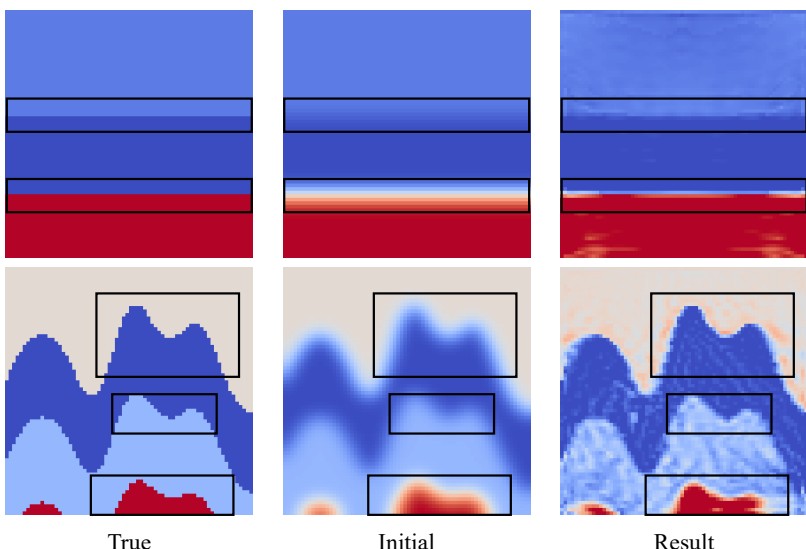

True                                 Initial                                 Result

Figure 12: Inconsistency between SSIM and recovered inversion details. For the top figure, the SSIM drops from an initial 0.732 to 0.508 (0.224↓). Similarly, the SSIM for the bottom figure decreases from an initial 0.608 to 0.529 (0.079 ↓) .

# H    GRADIENT VISUALIZATION

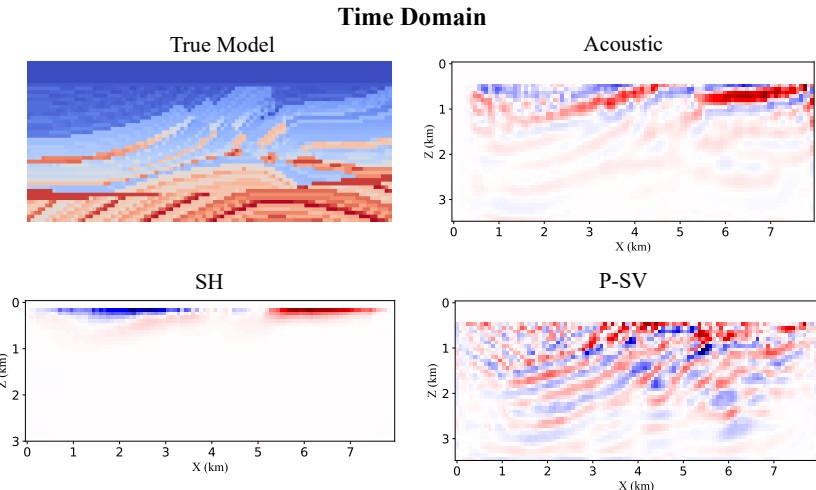

Figure 13: Gradient visualization of time-domain imaging.

**Frequency Domain**

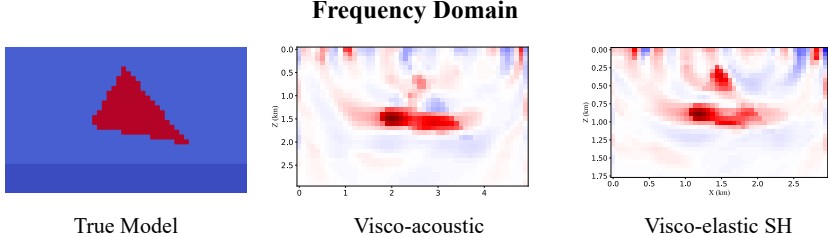

| True Model | Visco-acoustic | Visco-elastic SH |

Figure 14: Gradient visualization of frequency-domain Q imaging.

**Checkerboard Test**

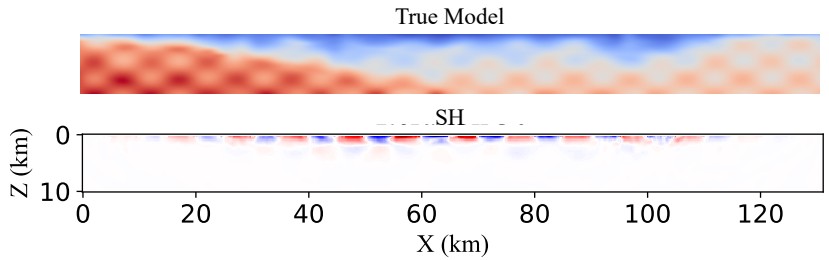

Figure 15: Gradient visualization of the field checkerboard test.

# I COST ANALYSIS

In order to conduct a comprehensive cost analysis that covers as many wave types, loss functions, and domains as possible, we consider two representative cases:

- acoustic wave with $L_2$ misfit in the time domain
- SH wave with amplitude misfit for attenuation in the frequency domain

## I.1 DERIVATION COST

The adjoint method requires challenging analytical derivations for each specific set of parameters, wave types, and loss functions. This process often involves tedious and difficult manual work, especially when extending to complex numerical computations.

However, no analytical derivation is required for our AD method.

### I.1.1 ACOUSTIC WAVE WITH $L_2$ MISFIT IN THE TIME DOMAIN

**Adjoint method** Adjoint source:

$$f^\dagger(\mathbf{x}_r, t) = d_{\text{syn}}(\mathbf{x}_r, t) - d_{\text{obs}}(\mathbf{x}_r, t)$$

Adjoint wavefield:

$$\frac{1}{c^2(\mathbf{x})} \frac{\partial^2 v(\mathbf{x}, t)}{\partial t^2} - \nabla^2 v(\mathbf{x}, t) = f^\dagger(\mathbf{x}, t)$$

Gradient:

$$\frac{\delta J}{\delta c(\mathbf{x})} = \frac{2}{c(\mathbf{x})} \int_0^T v(\mathbf{x}, t) \cdot \left(\nabla^2 u(\mathbf{x}, t)\right) dt$$

**AD Method** No analytical derivation is required.

### I.1.2 SH WAVE WITH AMPLITUDE MISFIT FOR ATTENUATION IN THE FREQUENCY DOMAIN

**Adjoint method**    Forward model:

$$Au = b, \quad v_s^{\text{complex}} = v_s + i \cdot \frac{v_s}{2Q}, \quad \mu^{\text{complex}} = \rho \cdot \left(v_s^{\text{complex}}\right)^2$$

Amplitude Misfit:

$$J = \sum_i \big| \log |d_i^{\text{obs}}| - \log |d_i^{\text{syn}}| \big|^2$$

Adjoint wavefield:

$$f^{\text{adj}} = -\operatorname{sign}\!\Big(\log |d_i^{\text{obs}}| - \log |d_i^{\text{syn}}|\Big) \cdot \frac{d_i^{\text{syn}}}{|d_i^{\text{syn}}|^2}, \qquad A^H \lambda = f^{\text{adj}}$$

Gradient:

$$\frac{\partial J}{\partial Q(i,j)} = \operatorname{Re}\left[ \lambda^*(i,j) \cdot \left(-\frac{\omega^2}{\left(\mu^{\text{complex}}\right)^2}\right) \cdot \frac{\partial \mu^{\text{complex}}}{\partial Q} \cdot u(i,j) \right]$$

where

$$\frac{\partial \mu^{\text{complex}}}{\partial Q} = \rho \cdot 2 v_s^{\text{complex}} \cdot \left(-i \frac{v_s}{2Q^2}\right)$$

Such adjoint derivations are tedious and error-prone, particularly when complex numbers are involved.

**AD Method**    No analytical derivation is required.

### I.2 IMPLEMENTATION COST

For each forward model and misfit, the adjoint approach requires separate backward solver implementation . Even for the simple $L_2$ misfit this means coding a dedicated time-reversal solver, while more advanced cases (e.g. amplitude misfit with attenuation) become non-self-adjoint and complex-valued.

**With AD, none of this is needed.**  Gradients are obtained directly by a single line of code: `loss.backward()`, and are theoretically and numerically exact.

**Workload comparison:** For $m$ forward models and $n$ misfits,

- Shared workload (easy):

$$m \text{ forward modeling } + \ n \text{ misfit implementations}$$

  As shared workload, $m$ forward simulation and $n$ misfit implementations are excluded from comparison.

- Workload saved by AD (challenging):

$$m \times n \text{ adjoint source derivations } + \ m \times n \text{ adjoint implementations}$$

AD thus saves the most challenging part, while supporting arbitrary wave equations and misfits in time and frequency domains.

### I.3 TIME AND MEMORY COST

Since deriving the adjoint wavefield with high-precision simulations is very challenging, we use simple simulations for testing here.

### I.3.1 ACOUSTIC WAVE WITH $L_2$ MISFIT (TIME DOMAIN, 10,000 TIME STEPS)

Table 19: Memory cost comparison .

| Size | AD | Adjoint | AD / Adjoint |
|---|---|---|---|
| 30×30×10000 | 0.15 GB | 0.08 GB | 1.88 |
| 100×100×10000 | 1.53 GB | 0.78 GB | 1.96 |
| 300×300×10000 | 13.53 GB | 6.81 GB | 1.99 |

Table 20: Time cost comparison .

| Size | AD | Adjoint | AD / Adjoint |
|---|---|---|---|
| 30×30×10000 | 4.2709 s | 2.7517 s | 1.55 |
| 100×100×10000 | 4.3207 s | 2.7518 s | 1.57 |
| 300×300×10000 | 5.0545 s | 2.9544 s | 1.71 |

### I.3.2 SH WAVE WITH AMPLITUDE MISFIT FOR ATTENUATION IN THE FREQUENCY DOMAIN

Table 21: Memory cost comparison.

| Size | AD | Adjoint | AD / Adjoint |
|---|---|---|---|
| 100×100 | 7.6 MB | 6.1 MB | 1.25 |
| 500×500 | 196.11 MB | 131.99 MB | 1.49 |

Table 22: Time cost comparison .

| Size | AD | Adjoint | AD / Adjoint |
|---|---|---|---|
| 100×100 | 1.3454 s | 1.0204 s | 1.32 |
| 500×500 | 41.4316 s | 32.3664 s | 1.28 |

## J  INITIAL MODEL DEPENDENCY

Our physics-driven method approach requires updating from an initial guess, which can usually be converted using ray-theory inversion or other seismic models.

Here we show the initial model dependencies on the OpenFWI dataset. Despite the observed initial model dependency (the higher the initial SSIM, the higher the resulting SSIM), our method demonstrates robustness to the quality of the initial model. For example, even when starting with a very blurred initial model (SSIM is only 0.4), it can still basically invert the model and capture the details.

In the future we explore incorporating deep learning methods to mitigate the reliance on initial models.

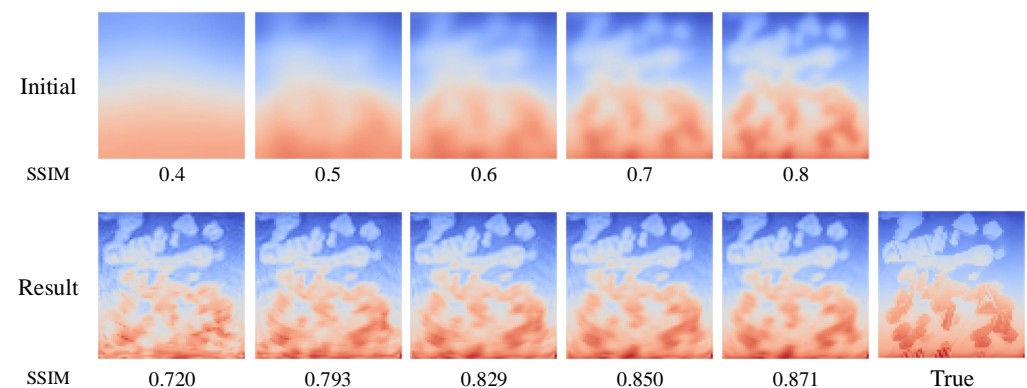

Figure 16: Initial model dependency.

## K  CROSS-SCENARIO RESULTS

Table 23: Cross-scenario results. **MAE↓ / MSE↓ / RMSE↓**

| Time Domain | FWI $L_2$ | Travel Time |
|---|---|---|
| Acoustic | 0.0401 / 0.0047 / 0.0688 | 0.0701 / 0.0117 / 0.1084 |
| SH | 0.0642 / 0.0095 / 0.0972 | 0.0645 / 0.0095 / 0.0975 |
| P-SV | 0.0723 / 0.0117 / 0.1080 | 0.0736 / 0.0148 / 0.1155 |
| **Frequency Domain** | **FWI $L_2$** | **Attenuation** |
| Visco-acoustic | 0.1034 / 0.0511 / 0.2260 | 0.1154 / 0.0647 / 0.2444 |
| Visco-elastic | 0.0927 / 0.0224 / 0.1495 | 0.0971 / 0.0219 / 0.1481 |

## L  ROBUSTNESS TEST

We evaluate the robustness of our AD-based framework under two challenging settings: varying noise levels and missing seismic traces.

As shown in Table 24, the method maintains basic performance even at a high noise level of 1% (SNR 10.27 dB), demonstrating strong noise resilience. Table 25 further shows that the reconstruction quality degrades gracefully, remaining acceptable even with 90% missing traces.

These results confirm the robustness and practicality of our framework for real-world seismic imaging.

Table 24: Performance under different noise levels.

| Noise level | 0% | 0.1% | 0.3% | 0.5% | 0.7% | 1% |
|---|---|---|---|---|---|---|
| SNR (dB) | – | 30.27 | 20.71 | 16.29 | 13.37 | 10.27 |
| SSIM | 0.7820 | 0.7322 | 0.6567 | 0.6086 | 0.6000 | 0.5966 |
| MAE | 0.0314 | 0.0358 | 0.0428 | 0.0458 | 0.0474 | 0.0460 |
| RMSE | 0.0443 | 0.0509 | 0.0571 | 0.0609 | 0.0630 | 0.0614 |

Table 25: Performance under different percentages of missing traces.

| Missing (%) | 0% | 1% | 4% | 10% | 20% | 50% | 70% | 90% |
|---|---|---|---|---|---|---|---|---|
| SSIM | 0.7820 | 0.7793 | 0.7808 | 0.7779 | 0.7806 | 0.7754 | 0.7551 | 0.6692 |
| MAE | 0.0314 | 0.0315 | 0.0314 | 0.0316 | 0.0317 | 0.0324 | 0.0330 | 0.0421 |
| RMSE | 0.0443 | 0.0448 | 0.0445 | 0.0446 | 0.0441 | 0.0453 | 0.0461 | 0.0562 |

