# OpenReview forum: "Unified, Practical, and White-box Seismic Tomography with Automatic Differentiation"
_ICLR.cc/2026/Conference — Submitted to ICLR 2026_

### Official Review · Reviewer_YP9p · 2025-10-17

**Soundness:** 4
**Presentation:** 4
**Contribution:** 3
**Rating:** 8
**Confidence:** 3

**Summary:**

Theoretically proves the equivalence of gradients between AD methods and
adjoint methods under frequency domain and any misfit functions. Numerically, defends its argument by experiments on different PDEs and wave types.

**Strengths:**

1. Extends AD methods to frequency domain with theoretical
justifications and numerical results. Previous AD methods are mostly restricted
to time domain for complexity reasons, while this work has a mathematical
proof and has achived unified results between time and frequency domain over
different problems.

2. Proofs and experiments are strong with clear explanations and straightforward
 diagrams.

3.  Inversion problems are notoriously hard, but this work has demonstrated the
 potential of AD methods with strong evidences.

**Weaknesses:**

Fails to demonstrate why diverse misfit functions are a difficulty.
 The challenges are neither discussed in mathematical deductions, nor further
 explained in experiments.

**Questions:**

1. In the context of AD, is implementing diverse misfit functions too trivial to be a significant contribution of this paper? AD-based methods were invented at least 4 years ago (ADSeismic) and people may have already been well aware of its flexibility compared to adjoint methods. Hence implementing diverse misfit functions may not be a significant contribution. (Or there do exist some particularly difficult misfit functions, but this paper does not discuss them in detail.)

2. Time domain methods are iteration based, and thus errors get maginified and divergence may happen. What is the time sample-rate  of the involved datasets? How do you address this problem?
A pure time domain method, especially over hundreds of steps, may be extremely unstable. Your frequency-based method may be a possible and robust solution.

3. How is the forward process of PDEs in frequency domain implemented? This might be worth mentioning since its not a trivial task.

---

> ### Author Response · Authors · 2025-11-20
>
> Thank you very much for your appreciation and constructive feedback! Many of your comments are truly insightful — they not only helped us improve the quality of our manuscript but also provided us with valuable ideas.
>
> > **Q1**: In the context of AD, is implementing diverse misfit functions too trivial to be a significant contribution of this paper? AD-based methods were invented at least 4 years ago (ADSeismic) and people may have already been well aware of its flexibility compared to adjoint methods. Hence implementing diverse misfit functions may not be a significant contribution. (Or there do exist some particularly difficult misfit functions, but this paper does not discuss them in detail.)
> >
> > **W1**: Fails to demonstrate why diverse misfit functions are a difficulty. The challenges are neither discussed in mathematical deductions, nor further explained in experiments.
>
> Thank you for these thoughtful comments! We respond to **Q1 and W1 together** because both comments concern the same core issues:
>
> (1) *why diverse misfit functions are a difficulty?*
>
> (2) *is implementing diverse misfit functions a significant contribution of this paper?*
>
>
>
> **(1) *why diverse misfit functions are a difficulty?***
>
> We fully agree with your premise that, in principle, AD offers flexibility for implementing diverse misfit functions, especially when compared to the traditional adjoint method. However, there remains a large gap between  ***' be well aware of AD’s flexibility '* as you mentioned** and ***' be able to conveniently and reliably use it in practice '* .**
>
> Without our framework, even if researchers are already aware of AD’s flexibility, implementing a new misfit function would still require them to:
>
> 1. Build a seismic imaging framework in an AD‑enabled language (e.g., PyTorch), including the basic gradient descent procedure, layout settings, etc.;
> 2. Implement the forward wave equation solver in that language adapted to the required misfit function (the most time‑consuming step);
> 3. Implement the misfit function itself within the same framework.
>
>  If multiple misfit functions are needed, **steps (2) and (3) would have to be repeated each time**.
>
> **In our framework, we have done all this heavy lifting in advance.**
>
>
>
> **(2) *is implementing diverse misfit functions a significant contribution of this paper?***
>
> We respectfully believe that **it is indeed a significant contribution**.
>
> As we mentioned in (1),  there remains a large gap between  *' be well aware of AD’s flexibility '* as you mentioned and *' be able to conveniently and reliably use it in practice '*. **Our work precisely bridges this gap in the aspects below:**
>
> 1. Make AD a **reliable** method. We **theoretically and numerically prove** that gradients from AD are equivalent to those from the analytical adjoint method, a proof that holds regardless of the domains, wave equations, or choice of misfit functions.
> 2. Make AD a **practical and versatile** method. We developed a customizable PyTorch-based AD seismic imaging framework across domains (time/frequency), waves (acoustic/SH/P-SV/visco-acoustic/visco-elastic), and misfit functions (waveform/travel time/amplitude). Our framework **enables users to effortlessly implement various misfit functions**.
>
>
>
> Thank you again for your question and for giving us the opportunity to further clarify our contributions!

---

> ### Author Response · Authors · 2025-11-20
>
> > **Q2**: Time domain methods are iteration based, and thus errors get maginified and divergence may happen. What is the time sample-rate of the involved datasets? How do you address this problem? A pure time domain method, especially over hundreds of steps, may be extremely unstable. Your frequency-based method may be a possible and robust solution.
>
> Thank you for this very insightful and valuable question!
>
> **Sampling rate:**
>
> (revised in E.3 CROSS-SCENARIO EXPERIMENTS, E.4 OPENFWI BENCHMARK EXPERIMENTS and E.5 FIELD EXPERIMENT):
>
> - Cross-scenario experiment
>   - acoustic wave: 0.006s
>   - SH / P-SV: 0.004s
> - OpenFWI
>   - Vel-B, Fault-B and Style-B Family: 0.001s
>   - Kimberlina-CO2: 0.002s
> - Nankai field experiment: 0.01s
>
>
>
> **Stability Problem: CFL Condition**
>
> Yes, in the time domain, inappropriate inverted parameters can lead to error magnification and divergence. To fundamentally address this issue, all our time-domain simulations strictly adhere to the **Courant-Friedrichs-Lewy (CFL) stability condition**.
>
> The CFL condition is a necessary condition for the convergence of explicit finite-difference schemes used to solve hyperbolic PDEs, such as the wave equation. The mathematical expression is typically:
>
> $$dt ≤ C \times (dx / v_{max})$$
>
> Where:
>
> - dt is the time step.
> - dx is the spatial grid spacing.
> - $v_{max}$ is the maximum wave velocity in the model.
> - C is the Courant number (often $\le$ 1, here we select 0.5 as a safe choice)
>
> In our implementation, dt is **not** chosen arbitrarily but is carefully calculated based on the grid spacing dx and the maximum velocity $v_{max}$  for each experimental model to ensure the CFL condition is always satisfied. During the inversion process, we continuously check whether the current updated model satisfies the CFL condition. If it does not, gradient clipping is applied to ensure that the CFL condition is maintained as a **priority**.
>
> The successful inversion results across numerous synthetic and field scenarios presented in the paper also prove our time-domain inversion's stability.
>
> We have added and highlighted this part in **Appendix D.1.1 TIME DOMAIN.**
>
>
>
>
>
> **Comparison with Frequency-Domain Methods:**
>
> Thank you for pointing out this major advantage of frequency‑domain methods.
> Yes, frequency‑domain methods are generally **more stable** than time‑domain ones. They solve the Helmholtz equations at different frequencies, which are **steady‑state** by nature. In contrast to time‑domain methods, frequency‑domain approaches **do not need to satisfy any CFL condition**.
>
> We will add this advantage in the revised manuscript.
>
> This observation highlights one of the core contributions of our work—**a unified framework**. The power and practicality of our platform lie precisely in its **integration of both time- and frequency-domain solvers**. This allows researchers to easily experiment with both approaches and obtain the best results.
>
>
>
> Thank you again for your valuable feedback, which helps us to further improve the clarity of our manuscript!
>
>
>
> > **Q3**: How is the forward process of PDEs in frequency domain implemented? This might be worth mentioning since its not a trivial task.
>
> Thank you for raising this excellent point about the implementation of our frequency-domain forward modeling. Yes, this is a non-trivial task, and we appreciate the opportunity to elaborate on our methodology.
>
> Our frequency-domain forward process is implemented using the **Frequency-Domain Finite-Difference (FDFD)** method. Specifically, for each frequency$\omega$ , a Helmholtz equation of the form $A_\omega u_\omega = s_\omega$ is solved. The key steps:
>
> 1. To Frequency-Dependent Complex Properties:
>
>    The simulation is based on the Helmholtz equation, which is the frequency-domain representation of the wave equation. To realistically model wave propagation in subsurface media, we incorporate attenuation effects(e.g., constant Q KF model, where material properties like velocity and bulk modulus become complex and frequency-dependent).
>
>    Specifically, in the KF model, the complex velocity is $\frac{1}{v(\omega)} = \frac{1}{v} + \frac{1}{\pi\,v\,Q}\,\ln\!\left(\frac{\omega_{\mathrm{ref}}}{\omega}\right) + \frac{i}{2\,v\,Q}$ and complex bulk modulus is $M(\omega) = \rho \cdot [v(\omega)]^2$.
>
> 2. Constructing $A_\omega$:
>
>    - Our method constructs $A_\omega$ as a sparse, complex-valued impedance matrix.
>
>    - Interior: **9-point finite-difference stencil** to minimize numerical dispersion
>
>    - Boundary Condition: Neumann condition for the free surface and PMLs for the other three boundaries, all implemented via **complex-stretching coordinates**
>
> 3. Solving the Large Sparse Linear System:
>
>    By setting the source term $ s_\omega$, the equation $A_\omega u_\omega = s_\omega$  can then be solved. Here, we use PyTorch’s built‑in solver.
>
>
>
> Thank you again for pointing this out! We have added and **highlighted this content in Appendix D.1.2 FREQUENCY DOMAIN.**

---

### Official Review · Reviewer_4MRU · 2025-10-30

**Soundness:** 2
**Presentation:** 2
**Contribution:** 2
**Rating:** 2
**Confidence:** 3

**Summary:**

This paper presents a framework for performing Full Waveform Inversion (FWI) through automatic differentiation (AD), implemented in PyTorch. Instead of deriving gradients via the traditional adjoint state method, the authors construct differentiable forward wavefield solvers and use PyTorch's AD mechanism to compute gradients of the loss. They theoretically and numerically demonstrate AD's effectiveness by proving that the gradients from AD are equivalent to those from the adjoint method. Then they demonstrate a few proof-of-concept inversions on synthetic and field examples.

**Strengths:**

1. They theoretically and numerically demonstrate AD's effectiveness by proving that the gradients
from AD are equivalent to those from the analytical adjoint method
2. Potential engineering contribution for an open-source package for various differentiable forward operators.

**Weaknesses:**

1. Unclear novelty and contribution. The paper does not propose a new FWI algorithm or learning method; it merely replaces the adjoint state gradient computation with PyTorch's automatic differentiation. Once the forward modeling is implemented in PyTorch, gradient computation follows directly from built-in autodiff; thus, the contribution is quite limited.

2. Writing and presentation need improvement: 1) The target problem and motivation are not clearly expressed. 2) Jargon terms and mathematical notation are used without adequate definition. 3) The underlying physical concepts are not explained. 4) The optimization objective of FWI is never formally stated, i.e., only gradient computation is discussed, which makes the end goal ambiguous.

3. Baseline results are not shown in the paper.

4. As shown in Figure 4, the inversion starts from a relatively strong initial, which reduces the challenge and overstates the effectiveness of the approach.

**Questions:**

Please refer to the weakness.

---

> ### Author Response · Authors · 2025-11-20
>
> Thank you for all the comments! Your insightful feedback helps us to improve the quality of the manuscript.
>
> > **W1**: Unclear novelty and contribution ...
>
> Thank you!  We would like to respectfully clarify that our work goes far beyond *“it merely replaces the adjoint state gradient computation with PyTorch's automatic differentiation”*.
>
>
>
> Yes, AD itself is not a fundamentally new technology; rather, we systematically analyze existing approaches (the **adjoint method and neural networks**) and identify several fundamental limitations—**practicality, interpretability, and fragmentation**. Through this analysis, we reveal that **straightforward AD—previously used merely as a substitute for adjoint gradients—is far more powerful than previously recognized and can effectively address these fundamental issues.**
>
> Building on this insight, we abandoned traditional FWI / prevailing neural networks and developed this **lower-level AD framework on a practical level** accompanied by **detailed theoretical and experimental analysis**.
>
>
>
> We respectfully show the **fundamental limitations** of existing approaches and **how we address them by the AD** framework:
>
> **1. Traditional adjoint method:**
>
> Traditional adjoint approaches are case-specific, requiring challenging analytical derivations for each set of parameters, waves, and loss functions:
>
> |                           | Adjoint Method                                   | **Our AD-based Framework** |
> | :------------------------ | :----------------------------------------------- | :------------------------- |
> | **Diverse Method**        | Fragmented (case-specific)                       | **Unified**                |
> | **Analytical Derivation** | $m \times n$ adjoint sources and wavefields      | **None**                   |
> | **Implementation**        | $m \times n$ time-reversal solvers and operators | **None**                   |
>
>
>
> Our contributions regarding the adjoint method:
>
> - We **theoretically and numerically proved** that gradients from AD are equivalent to those from the analytical adjoint method, regardless of the domain, wave equation, or choice of misfit function.
> - We developed a customizable PyTorch-based AD seismic imaging framework, the **first to concurrently support both time and frequency domains**. This design allows it to be directly used or embedded into deep learning workflows.
> - We comprehensively validated our framework through experiments across ten diverse scenarios.

---

> ### Author Response · Authors · 2025-11-20
>
> **2.  NN‑based methods** (e.g., **PINNs** and **neural operators**)
>
> **More importantly**, our work rethinks and addresses the **fundamental limitations of NN-based methods**, particularly their **limited practicality and poor interpretability** when applied to real-world problems:
>
> |                          | **NN-based Methods (e.g., PINNs)**                           | **Our AD-based Framework**                                   |
> | :----------------------- | :----------------------------------------------------------- | :----------------------------------------------------------- |
> | **Data Requirements**    | Observed data; Prohibitively large dataset for practical use (e.g., various geology settings, survey geometry settings) | Observed data; A rough initial model. (**universally applicable** to any geology and geometry ) |
> | **Detail Recovery**      | Smoothed, potentially missing key anomalies.                 | Good at recovering high-frequency details.                   |
> | **Physical Consistency** | Enforced via a loss term, but **not guaranteed** in the final result. | **Guaranteed**.                                              |
> | **Interpretability**     | **Black-box/Grey-box**.                                      | **Completely white-box**. Every step is physically and mathematically meaningful. |
> | **Practicality**         | Limited.                                                     | **High (tested in Nankai subduction zone)**.                 |
>
> The **Nankai subduction zone** experiment in our paper is a perfect case study. An NN-based approach is impractical here because no dataset is suitable:
>
> 1. **Structural Mismatch:** The geology of a subduction zone is structurally different from the models in existing datasets (e.g., OpenFWI).
>
> 2. **Scale and Geometry Mismatch:** The scale and layout of the Nankai survey ( **131 × 10 km**, **5 km** source spacing) are much larger than OpenFWI (**0.7 × 0.7 km, 0.14 km** source spacing).
>
>
>
> Our contributions regarding NN-based methods:
>
> - We **abandoned neural networks** and instead built a **fully white‑box and practical seismic imaging framework** based on **lower‑level AD**.
>
> - We conducted **comprehensive comparisons on the OpenFWI dataset**, demonstrating that our method achieves the best performance on the  Style-B subset with the most high-frequency structures.
>
> - We successfully applied our method to the **Nankai subduction zone**, a challenging real-world case **where purely NN-based methods are difficult to apply**, proving the robustness and practical superiority of our framework.
>
>
>
> Finally, regarding your comment that “*once the forward modeling is implemented in PyTorch, gradient computation follows directly from built‑in autodiff*”, this is **exactly the strength of our framework** — one only needs to implement the **forward modeling**. Especially in real‑world applications, our approach avoids both the **prohibitive training data requirements** of learning‑based methods and the **repeated analytical derivations** required by traditional adjoint‑state methods.
>
>
>
> Thanks so much for your valuable and insightful feedback!
>
>
>
>
>
>
> > **W2 - 1)** Writing and presentation need improvement: 1) The target problem and motivation are not clearly expressed.
>
> Thank you. This part has been **revised and highlighted in the Introduction**; please refer to our **response to W1** for details.
>
> > **W2 - 2)** Jargon terms and mathematical notation are used without adequate definition.
>
> Thank you. We have carefully checked and added jargon terms and mathematical notation.We have revised and highlighted them in the manuscript:  **2 PROBLEM SETUP and 5.1 THEORETICAL PROOF.**
>
> > **W2 - 3)**  The underlying physical concepts are not explained.
>
> Thank you. We have added in **Appendix D. FORWARD AND BACKWARD MODULES.**
>
> > **W2 - 4)** The optimization objective of FWI is never formally stated, i.e., only gradient computation is discussed, which makes the end goal ambiguous.
>
> Thank you. This part has been **revised and highlighted in the first paragraph in Introduction**:
>
> This approach inverts the model by minimizing the seismic data–simulation misfit with the computed gradient, *i.e.*,  it seeks the model
>
> $$
> \mathbf{m}^* = \arg\min_{\mathbf{m}} J\left(\mathbf{d}_{\text{obs}}, F(\mathbf{m})\right),
> $$
>
> where$ J (\cdot,\cdot) $denotes the misfit function measuring the difference between the observed data $\mathbf{d}_{\text{obs}}$ and the simulated data $ F(\mathbf{m})$.

---

> ### Author Response · Authors · 2025-11-20
>
> > **W3**: Baseline results are not shown in the paper.
>
> Thank you for your comment.
>
> If by *baseline results* you are referring to ADFWI (Liu et al., 2024) and the Matlab-based visco-acoustic wave equation solver (Amini & Javaherian, 2011), these are merely the ***baseline code frameworks*** used in our work. Our AD framework serves a different purpose from theirs. Therefore, **there is no *performance comparison* in the deep-learning sense** between our method and these baselines. In the revised manuscript, we have replaced the ambiguous term *“baseline”* with *“baseline codes”* to avoid misunderstanding and ensure clarity.
>
> If, however, by *baseline results* you are referring to the comparison with **state-of-the-art methods on the benchmark datasets**, please refer to the table below. (Compared with the table in the previous version, Reviewer 13jS suggested adding MAE and RMSE, as well as including the stronger method [1] for comparison.)
>
>
>
> | SSIM↑          | BigFWI-B |  BigFWI-M  |  BigFWI-L  | InversionNet | VelocityGAN | UPFWI  |  **Ours (init.)**   |
> | :------------- | :------: | :--------: | :--------: | :----------: | :---------: | :----: | :-----------------: |
> | FlatVel-B      |  0.9658  |   0.9729   | **0.9756** |    0.9356    |   0.9556    | 0.8874 |   0.5673 (0.6978)   |
> | CurveVel-B     |  0.7808  |   0.8053   | **0.8134** |    0.6630    |   0.7111    | 0.6614 |   0.5216 (0.6207)   |
> | FlatFault-B    |  0.8027  | **0.8137** |   0.8033   |    0.7323    |   0.7552    | 0.6937 |   0.6518 (0.6622)   |
> | CurveFault-B   |  0.6781  | **0.6896** |   0.6790   |    0.6137    |   0.6033    |   -    |   0.5762 (0.5772)   |
> | Style-B        |  0.7567  |   0.7600   |   0.7429   |    0.7667    |   0.7249    | 0.6102 | **0.8093** (0.5552) |
> | Kimberlina-CO₂ |    -     |     -      |     -      |  **0.9872**  |   0.9716    |   -    |   0.9489 (0.7945)   |
>
> | MAE↓           | BigFWI-B | BigFWI-M |  BigFWI-L  | InversionNet | VelocityGAN | UPFWI  |  **Ours (init.)**   |
> | :------------- | :------: | :------: | :--------: | :----------: | :---------: | :----: | :-----------------: |
> | FlatVel-B      |  0.0233  |  0.0193  | **0.0173** |    0.0304    |   0.0328    | 0.0677 |   0.0395 (0.0402)   |
> | CurveVel-B     |  0.0933  |  0.0816  | **0.0772** |    0.1448    |   0.1428    | 0.1777 |   0.0813 (0.0958)   |
> | FlatFault-B    |  0.0710  |  0.0636  |   0.0644   |    0.0965    |   0.0946    | 0.1416 | **0.0544** (0.0689) |
> | CurveFault-B   |  0.1245  |  0.1161  |   0.1169   |    0.1705    |   0.1583    | 0.3452 | **0.1034** (0.1313) |
> | Style-B        |  0.0553  |  0.0538  |   0.0563   |    0.0557    |   0.0649    | 0.1702 | **0.0399** (0.0757) |
> | Kimberlina-CO₂ |    -     |    -     |     -      |  **0.0061**  |   0.0119    |   -    |   0.0103 (0.0193)   |
>
> | RMSE↓          | BigFWI-B |  BigFWI-M  |  BigFWI-L  | InversionNet | VelocityGAN | UPFWI  |  **Ours (init.)**   |
> | :------------- | :------: | :--------: | :--------: | :----------: | :---------: | :----: | :-----------------: |
> | FlatVel-B      |  0.0696  |   0.0621   | **0.0584** |    0.0680    |   0.0787    | 0.1493 |   0.0718 (0.0879)   |
> | CurveVel-B     |  0.2154  |   0.2006   | **0.1947** |    0.3111    |   0.2611    | 0.3179 |   0.2073 (0.2251)   |
> | FlatFault-B    |  0.1321  | **0.1259** |   0.1269   |    0.1636    |   0.1553    | 0.2220 |   0.1283 (0.1398)   |
> | CurveFault-B   |  0.2027  |   0.1954   |   0.1960   |    0.2507    |   0.2336    | 0.5010 | **0.1918** (0.2050) |
> | Style-B        |  0.0876  |   0.0867   |   0.0908   |    0.0860    |   0.0979    | 0.2609 | **0.0406** (0.0921) |
> | Kimberlina-CO₂ |    -     |     -      |     -      |    0.0374    |   0.0387    |   -    | **0.0195** (0.0403) |
>
>
>
>
>
> [1] Jin, P., Feng, Y., Feng, S., Wang, H., Chen, Y., Consolvo, B., ... & Lin, Y. (2024). An empirical study of large-scale data-driven full waveform inversion. Scientific Reports, 14(1), 20034.

---

> ### Author Response · Authors · 2025-11-20
>
> > **W4**: As shown in Figure 4, the inversion starts from a relatively strong initial, which reduces the challenge and overstates the effectiveness of the approach.
>
> Thank you for your thoughtful question. We understand your concern: it may seem that inversion only works well if the initial model already contains basic features of the true structure.
>
> The need for a good initial model is a common challenge for **all physics-driven inversion methods**, including both the adjoint method and our AD-based method. This is because seismic inversion is an **ill-posed and non-unique** problem. There can be many different models that fit the observed data well, even though only one is correct in reality.
>
> | **Initial SSIM** | 0.4   | 0.5   | 0.6   | 0.7   | 0.8   |
> | ---------------- | ----- | ----- | ----- | ----- | ----- |
> | **Result SSIM**  | 0.720 | 0.793 | 0.829 | 0.850 | 0.871 |
>
> As shown in Figure 14, when the initial SSIM drops to **0.4 (where no structure is visually recognizable)**, our method **still reconstructs most model details**, with a final SSIM of 0.72. Even compared to the result from an initial SSIM of 0.8 (0.4 higher), the gap is only 0.151.
>
> **In the future, we plan to use deep learning to generate initial models**, as deep learning can map seismic recordings directly to subsurface models without requiring any prior information (basic features of the solution you mentioned). Although these results are coarse, they are generally accurate and thus provide suitable starting points for further refinement by our physics-driven approach.
>
> We appreciate your feedback and clarify this point in **Appendix J. INITIAL MODEL DEPENDENCY**.

---

> ### Comment · Reviewer_4MRU · 2025-11-27
>
> Thank you for the detailed response. However, my main concern is still unresolved.
>
> I remain confused about what the concrete contribution of the paper is. In your response, you provided an extended discussion of why automatic differentiation (AD) is useful, but you did not clearly explain what is actually new in your implementation, nor how your framework differs in practice from existing ADFWI.
>
> Since your own response states that "...this is exactly the strength of our framework — one only needs to implement the forward modeling," it remains unclear whether your practical contribution is simply re-implementing forward modeling operators in PyTorch. Mechanistically, both use AD to compute FWI gradients, and you mentioned that ADFWI is your baseline code. Or should I think of your work mainly as a theoretical proof of the effectiveness of AD rather than an algorithmic or methodological contribution?
>
> More importantly, since the work is a purely physics-based FWI paper with no machine learning component, I am unsure about its suitability for ICLR. The paper may be valuable for the broader FWI community, but I do not see why it is important for the ICLR audience, which typically focuses on machine learning methods, learning representation, or related topics. I would appreciate clarification on why the authors believe this paper fits the ICLR scope.

---

> > ### Author Response · Authors · 2025-11-27
> >
> > Thank you for the insightful feedback. We fully understand your concerns and we respectfully reply point by point.
> >
> >
> >
> > > I remain confused about what the concrete contribution of the paper is.  In your response, you provided an extended discussion of why automatic differentiation (AD) is useful, but you did not clearly explain what is actually new in your implementation, nor how your framework differs in practice from existing ADFWI ... Or should I think of your work mainly as a theoretical proof of the effectiveness of AD rather than an algorithmic or methodological contribution?
> >
> > In short, the core novelty of our method lies in **carefully evaluating the practicality issues of neural network (NN)–based seismic imaging**, and **proposing using low‑level AD to address these challenges.**  We show that AD is not merely a tool to avoid manual gradient derivation, as traditionally viewed, but also provides unifying capability, strong practicality, and high interpretability in inverse problems.
> >
> > Although both our paper and previous pure‑AD research (e.g., ADFWI) use AD, our work is **fundamentally different**:
> >
> > 1. **Motivation**
> >
> >    Our goal is to address **the *practicality issues* of NN–based seismic imaging**—specifically its **limited generalizability and interpretability**.
> >
> >    In contrast, the motivation of ADFWI is to simplify the computation of FWI gradients.
> >
> > 2. **Research domain**
> >
> >    Our work is within ***NN–based seismic imaging***, whereas ADFWI belongs to conventional **FWI**.
> >
> > Our paper includes extensive comparisons with the adjoint‑state FWI method (including proofs of equivalence), but this is **not** because our contribution lies in re‑deriving ADFWI. Instead, the purpose is to demonstrate that our **AD‑based method is effective for tackling *real‑world problems that traditional NN‑based imaging methods may fail to generalize*.**
> >
> >
> >
> > > ... the work is a purely physics-based FWI paper with no machine learning component ... I would appreciate clarification on why the authors believe this paper fits the ICLR scope.
> >
> > 1. **Our motivation originates directly from NN–based seismic imaging.**
> >
> >    Recent advances such as PINNs and neural operators have shown impressive performance. However, when we attempted to apply these NN-based approaches from academic benchmarks to realistic seismic imaging tasks, we consistently observed that **they were impractical due to limited generalizability and interpretability**. This challenge prompted us to revisit low‑level AD instead of relying on training‑based neural networks.
> >
> >    Therefore, our work is intended for the ICLR audience—especially researchers who aim to deploy learning methods in real‑world settings but currently face issues of data scarcity or generalization limits.
> >
> > 2. Forward wave propagation is an **analog recurrent neural network**, providing a form of **representation learning.**
> >
> >    Iteratively updating the propagation medium velocity is equivalent to training a recurrent neural network (RNN) [1]. This type of learning, called an analog machine‑learning hardware platform, promises to be faster and more energy‑efficient than its digital counterparts.
> >
> >    In this view, our framework performs ***representation learning governed by wave physics***: the subsurface model is the “network”, the input tensor is the wave source, and the output tensor is the waveform recorded at each receiver. In future work, we will further explore the potential of our approach as an analog recurrent neural network in domains beyond seismic imaging.
> >
> >    We will make this point much clearer in the revised manuscript.
> >
> > 3. **Benchmarking against ML methods**
> >
> >    We explicitly evaluate our method against deep‑learning methods (InversionNet, VelocityGAN, UPFWI, and BIGFWI) on the OpenFWI benchmark. We analyze where “pure physics” outperforms “pure DL” / “hybrid” methods (generalization, high‑frequency details) and where it fails (noise sensitivity). This analysis provides critical insights for the ICLR community.
> >
> >
> >
> > Thank you for your feedback again. We hope this clarifies the contribution of our work and why it is relevant to the ICLR community.
> >
> >
> >
> > [1] Hughes, Tyler W., et al. "Wave physics as an analog recurrent neural network." *Science advances* 5.12 (2019): eaay6946.

---

> ### Comment · Reviewer_4MRU · 2025-11-28
>
> Thank the authors for the detailed reply.
>
> However, several of issues remain unresolved:
>
> * You emphasize that your motivation differs from ADFWI, but you still do not explain what is actually different in practice.
> The response focuses on conceptual distinctions, but does not describe any concrete implementation or algorithmic differences between your framework and previous AD-based FWI systems such as ADFWI.
>
> * The claim of “carefully evaluating the practicality issues of NN-based seismic imaging” is overstated. A conceptual comparison table and a benchmark comparison without ablations or deeper analysis cannot be considered a careful or thorough evaluation.
>
> * The claim of “proposing low-level AD to address these challenges” is not novel. Using AD for FWI is neither conceptually new nor practically new.
>
> * Using AD or Pytorch does not make the method a NN-based approach. Similarly, iteratively updating a parameter does not make a method an RNN.
>
> * The limitations of NN-based methods (generalization, interpretability) are well-known.
> Presenting these shortcomings is not a new insight of this paper.
>
> Overall, the response does not address the questions regarding novelty, practical contribution, or the suitability for ICLR. My concerns therefore remain.

---

> > ### Author Response · Authors · 2025-11-28
> >
> > Thank you for all your comments.
> >
> > > ... what is actually different ...
> >
> > Please find the differences between previous AD works and ours:
> >
> > - ADFWI (Liu et al., 2024):
> >
> > |           | ADFWI          | Ours                                                         |
> > | --------- | -------------- | ------------------------------------------------------------ |
> > | Wave Type | Acoustic, P–SV | Acoustic, P–SV, **SH, Visco‑acoustic, Visco‑elastic**        |
> > | Domain    | Time           | Time, **Frequency**                                          |
> > | Misfit    | $L_2$ waveform | **Arbitrary, e.g., waveform, travel time, and attenuation...** |
> >
> > - ADseismic (Zhu et al. 2021):
> >
> > |                   | ADSeismic                       | Ours                                                         |
> > | ----------------- | ------------------------------- | ------------------------------------------------------------ |
> > | Equivalence Proof | Time‑domain forward + L2 misfit | Time‑ & **frequency‑domain** forward + **general misfit**    |
> > | Domain            | Time                            | Time & **frequency**                                         |
> > | Wave Types        | Acoustic, P‑SV                  | **General** functionals (Acoustic, SH, P‑SV, Visco‑acoustic, Visco‑elastic, etc.) |
> > | Misfit            | L2                              | **General** functionals (L2, travel time, attenuation, etc.) |
> > | Language          | Julia                           | Python (**PyTorch**)                                         |
> >
> >
> >
> > > ... “carefully evaluating the practicality issues of NN-based ...” is overstated...
> >
> > We did not overstate our contribution. We tested NN‑based methods on the Nankai subduction zone.
> >
> > A necessary condition for effective NN‑based seismic imaging is that the input Δx and Δt must match those in the training set.  However, because the scale of our task is considerably different from that of OpenFWI—and so large that it is difficult to feed into the network—we were unable to obtain **quantitative metrics** as traditional NN-based models do not work on the Nankai subduction zone dataset.
> >
> > 1. **Structural Mismatch:** The geology of a large-scale subduction zone is structurally different from the training datasets and smaller models in OpenFWI.
> >
> > 2. **Scale and Geometry Mismatch:** The scale and layout of the Nankai survey (**131 × 10 km**, 5 km source spacing) are much larger than OpenFWI (0.7 × 0.7 km, 0.14 km source spacing).
> >
> > Therefore, our comparison is presented as a conceptual comparison table. And we did not overstate our contribution.
> >
> >
> >
> > > ... Using AD for FWI is neither conceptually new nor practically new.
> >
> > We respectfully clarify again that our use of RNN and AD is not aimed at solving the traditional problems in FWI, but at addressing the limitations of NN‑based methods (generalizability and interpretability).
> >
> > To the best of our knowledge, AD has indeed not been used to address these limitations of NN‑based methods for wave physics in both time&frequency domains and various loss functions.
> >
> >
> >
> > > Using AD or Pytorch does not make the method a NN-based approach... iteratively updating a parameter does not make a method an RNN.
> >
> > We respectfully clarify that we did not claim our method is NN‑based (PINN or neural operators); rather, we stated that our method addresses the limitations of NN‑based methods (generalizability and interpretability).
> >
> > Second, we did not say “*iteratively updating a parameter does not make a method an RNN*.”  What we stated is: “*iteratively updating the propagation‑medium velocity is equivalent to training an RNN.*”  This is because the physics of time-dependent wave propagation and the forward pass of an RNN are highly similar in the  analog recurrent neural network literature (please refer to the well-known paper [1]).
> >
> >
> >
> > > The limitations ... are well-known...
> >
> > The limitations of NN‑based methods (generalization and interpretability) are well‑known.  However, with the increasing number of datasets, generalization has significantly improved (as in LLMs), and advances in theoretical research have also improved interpretability [2].
> >
> > But **in the field of seismic imaging, these issues remain unresolved for NN‑based methods**.
> >
> > Here we avoid the limited generalizability of NN‑based seismic imaging by combining wave physics with RNN and AD. Our use of RNN and AD is completely white‑box, and we provide **mathematical explanations for every update step**.  Since AD is a fundamental component of neural networks, we plan to extend our work to NN‑based approaches in the future to further study interpretability in NN‑based seismic imaging.
> >
> >
> >
> >
> >
> > Thank you, and we hope these responses address your concerns.
> >
> >
> >
> > [1] Hughes, Tyler W., et al. "Wave physics as an analog recurrent neural network." *Science advances* 5.12 (2019): eaay6946.
> >
> > [2] Yu, Yaodong, et al. "White-box transformers via sparse rate reduction." *Advances in Neural Information Processing Systems* 36 (2023): 9422-9457.

---

### Official Review · Reviewer_7KA1 · 2025-10-31

**Soundness:** 3
**Presentation:** 3
**Contribution:** 2
**Rating:** 2
**Confidence:** 3

**Summary:**

The paper proposes a unified framework for seismic tomography using automatic differentiation (AD) for gradient computation, aiming to avoid the manual derivation required by traditional adjoint methods. The authors claim their AD-based approach is theoretically and numerically equivalent to the adjoint method, supports a wide range of wave equations and misfit functions, and is practical for real-world scenarios. They provide a PyTorch-based open-source platform and validate their method across synthetic, benchmark, and field datasets.

**Strengths:**

1. Unified Framework: The paper presents a generalizable approach that can handle various wave types, domains (time/frequency), and misfit functions, which is broader than previous AD-based works.
2. Theoretical and Numerical Equivalence: The authors provide detailed proofs and numerical experiments showing that AD gradients match those from the adjoint method.
3. Open-Source Implementation: The release of a PyTorch-based platform may benefit the community and facilitate further research.
Practical Validation: The method is tested on synthetic models, the OpenFWI benchmark, and a field experiment, demonstrating applicability.

**Weaknesses:**

1. Lack of Deep Insight or Novelty: The main contribution is the application of AD to seismic tomography, but this is not fundamentally new. The equivalence between AD and adjoint gradients is well-understood in the scientific computing community, and the paper itself cites prior works that have already used AD for integrating physics into inverse problems .
The theoretical proof of equivalence is a straightforward application of chain rule, which is standard in optimization and inverse problems.
The practical advantage (avoiding manual derivation) is incremental rather than transformative, as modern frameworks (e.g., PyTorch, JAX) already make AD accessible.
2. Comparison to Conventional Gradient Computation:
The paper claims AD gradients are equivalent to adjoint gradients, which is correct. If AD does not match the conventional adjoint method, either implementation is wrong. This is a well-established fact in computational physics and inverse problems.

**Questions:**

Beside the implementation and showing the equivalence of AD and conventional method, what else the main contribution of this paper? AD can handle different misfit functions, and practically we also observed the success of using misfit function beyond l2 loss, e.g., Unsupervised learning of full-waveform inversion: Connecting cnn and partial differential equation in citation.

---

> ### Author Response · Authors · 2025-11-20
>
> Thank you very much for all your insightful comments!
>
>
>
> > **W1**: Lack of Deep Insight or Novelty ...
>
> Thank you for your comments!
>
>
>
> Yes, AD itself is not a fundamentally new technology; rather, we systematically analyze existing approaches (the **adjoint method and neural networks**) and identify several fundamental limitations—**practicality, interpretability, and fragmentation**. Through this analysis, we reveal that **straightforward AD—previously used merely as a substitute for adjoint gradients—is far more powerful than previously recognized and can effectively address these fundamental issues.**
>
> Building on this insight, we abandoned traditional FWI / prevailing neural networks and developed this **lower-level AD framework on a practical level** accompanied by **detailed theoretical and experimental analysis**.
>
>
>
> In addition, regarding your comment that *“The equivalence between AD and adjoint gradients is well-understood ...”* we **respectfully clarified this point in our response to Comment W2.**
>
>
>
> We respectfully show the **fundamental limitations** of existing approaches and **how we address them by AD** framework:
>
> **1. Traditional adjoint method:**
>
> |                           | Adjoint Method                                   | **Our AD-based Framework** |
> | :------------------------ | :----------------------------------------------- | :------------------------- |
> | **Diverse Method**        | Case-specific                                    | **Unified**                |
> | **Analytical Derivation** | $m \times n$ adjoint sources and wavefields      | **None**                   |
> | **Implementation**        | $m \times n$ time-reversal solvers and operators | **None**                   |
>
> Our contributions regarding the traditional adjoint method:
>
> - We **theoretically and numerically proved** that gradients from AD are equivalent to those from the analytical adjoint method, regardless of the domain, wave equation, or choice of misfit function.
> - We developed a customizable PyTorch-based AD seismic imaging framework, the **first to concurrently support both time and frequency domains**. This design allows it to be directly used or embedded into deep learning workflows.
> - We comprehensively validated our framework through experiments across ten diverse scenarios.
>
>
>
> **2.  NN‑based methods** (e.g., **PINNs** and **neural operators**)
>
> **More importantly**, our work rethinks and addresses the **fundamental limitations of NN-based methods**, particularly their **limited Generalizability and interpretability** when applied to real-world problems:
>
> |                          | **NN-based Methods (e.g., PINNs)**                           | **Our AD-based Framework**                                   |
> | :----------------------- | :----------------------------------------------------------- | :----------------------------------------------------------- |
> | **Data Requirements**    | Observed data; Prohibitively large dataset for practical use (e.g., various geology settings, survey geometry settings) | Observed data; A rough initial model. (**Universally applicable** to any geology and geometry ) |
> | **Detail Recovery**      | Smoothed, potentially missing key anomalies.                 | Good at recovering high-frequency details.                   |
> | **Physical Consistency** | Enforced via a loss term, but **not guaranteed** in the final result. | **Guaranteed**.                                              |
> | **Interpretability**     | **Black-box/Grey-box**.                                      | **Completely white-box**. Every step is physically and mathematically meaningful. |
> | **Generalizability**     | Limited.                                                     | **High (tested in Nankai subduction zone)**.                 |
>
> Our contributions regarding NN-based methods:
>
> - We **abandoned neural networks** and instead built a **fully white‑box and practical seismic imaging framework** based on **lower‑level AD**.
>
> - We conducted **comprehensive comparisons on the OpenFWI dataset**, demonstrating that our method achieves the best performance on the  Style-B subset with the most high-frequency structures.
>
> - We successfully applied our method to the **Nankai subduction zone**, a challenging real-world case **where purely NN-based methods are difficult to apply**, proving the robustness and practical superiority of our framework.
>
>
>
> Thank you for your insightful comment! We have revised and **highlighted in Abstract and Introduction.**

---

> > ### Author Response · Authors · 2025-11-20
> >
> > > **W2**: Comparison to Conventional Gradient Computation: The paper claims AD gradients are equivalent to adjoint gradients, which is correct. If AD does not match the conventional adjoint method, either implementation is wrong. This is a well-established fact in computational physics and inverse problems.
> >
> > Thank you for your valuable feedback. We agree that, at a high level, the principle underlying back-propagation is the same as that of the adjoint method and reverse-mode AD.
> >
> > However, when it comes to seismic imaging problems, **AD and the adjoint method cannot be assumed to be inherently equivalent**:
> >
> > - For completeness and unification, our work involves **complex‑valued operators** to study more complex subsurface media (e.g., energy attenuation). This introduces an entirely new computational regime that requires a special set of differentiation rules — **Wirtinger calculus**.
> >
> >   For example, compare the difference in differentiating the $L_2$ norm between the real‑valued and complex‑valued cases under Wirtinger calculus:
> >
> >   - Real-number case:  $f(x) = |x|^2$. The derivative is simply: $\frac{df}{dx} = 2x$.
> >
> >   - Complex-number case (Wirtinger calculus):   $f(z) = |z|^2$. The derivative is  the $\frac{df}{dz} = [z^*    z]^T$.
> >   - **Even the most common $L_2$ norm behaves completely differently, not to mention the far more complex wave‑equation modeling and other misfit functions.**
> >
> > - In the time domain, **the step‑by-step formulation** of the adjoint method differs significantly from the **frequently-used matrix formulation of AD**, so the high‑level equivalence does not always hold in practice.
> >
> > - Our work directly addresses **generalized misfit functionals**, whereas prior proofs in the seismic field were limited to the $L_2$ norm, which greatly simplifies the analysis but limits generality.
> >
> >
> >
> > To summarize, when it comes to specific seismic imaging problems, the adjoint method and AD are **not inherently equivalent**—such an assumption oversimplifies the underlying mathematics (including the step‑by‑step formulation, Wirtinger calculus, and generalized misfit functionals). Therefore, conducting implementation experiments to numerically verify their equivalence is **necessary and reasonable**.
> >
> >
> >
> > > **Q1**:Beside the implementation and showing the equivalence of AD and conventional method, what else the main contribution of this paper? AD can handle different misfit functions, and practically we also observed the success of using misfit function beyond l2 loss, e.g., Unsupervised learning of full-waveform inversion: Connecting cnn and partial differential equation in citation.
> >
> > Thank you for this insightful question.
> >
> > Besides our forthcoming open‑source practical AD framework and the theoretica/numerical demonstration of equivalence between AD and the adjoint method, our main contributions are:
> >
> > -  We show new insights into the **broader role of AD in inverse problems**. AD is not merely a tool to avoid manual gradient derivation, as traditionally viewed, but also provides strong practicality (tested in Nankai) and white‑box interpretability.
> > -  We systematically analyze the **fundamental limitations of existing NN-based methods**, particularly their **limited practicality and poor interpretability** when applied to real-world problems. These limitations indicate promising directions for future research.
> > -  We conducted comprehensive experiments on AD‑based seismic imaging, including **ten cross‑scenario experiments**, the **OpenFWI benchmark test**, and a **field experiment in the Nankai subduction zone**.
> >
> >
> >
> > Regarding **UPFWI** (*Unsupervised learning of full‑waveform inversion: Connecting CNN and partial differential equations*) you mentioned, we agree that UPFWI can also incorporate non‑L2 misfit functions within its unsupervised learning framework.
> >
> > However, UPFWI **remains neural‑network‑based**, and thus inherits the limitations we discussed in our  **W1 reply**— **limited interpretability (a gray‑box nature)** and a tendency toward **over‑smoothed inversion results**. In contrast, our approach is **fully white‑box and physics‑driven**. Moreover, in the OpenFWI benchmark experiments (Table 3,4,5), **our method surpasses UPFWI on structurally complex datasets (e.g., Style‑B)**, further supporting the practical advantages of our AD‑based framework.
> >
> >
> >
> > Thank you for your insightful comment!

---

### Official Review · Reviewer_13jS · 2025-11-01

**Soundness:** 2
**Presentation:** 3
**Contribution:** 2
**Rating:** 4
**Confidence:** 4

**Summary:**

This paper proposed a unified framework for seismic tomography that leverages automatic differential (AD) to compute gradients. This framework works on a case-by-case basis and can be applied across both time and frequency domains with various wave types and diverse misfit functions. A theoretical proof is provided by the authors to demonstrate the equivalence of the gradients of their framework and adjoint methods. Experiments show the effectiveness of the framework across various scenarios. Furthermore, the authors performed a cost analysis and highlighted the advantages of their method.

**Strengths:**

1. The paper is well-organized and easy to follow.
2. The authors provided detailed proof to theoretically demonstrate the equivalence of the gradients of the proposed framework and adjoint methods.
3. The experiments  covers 10 scenarios with different domains, wave types and misfit functions, offering a comprehensive evaluation that shows the effectiveness of the proposed framework.
4. The authors evaluated their proposed framework on the public benchmark dataset OpenFWI, which makes the experiments more convincing. Besides, a field experiment is conducted to further illustrate its potential for real-world applications.

**Weaknesses:**

1. The authors compared their proposed framework with adjoint method to show the advantages of AD. However, there also exist other methods that leverage AD such as Physics-Informed Neural Network (PINN), which also avoids analytical derivation. I think it is crucial to provide a comparison of the proposed framework and such methods. Otherwise, the novelty may be limited.
2. In the experiment section, the author stated that there are two baselines (ADFWI and a Matlab-based visco-acoustic wave equation solver). However, the performance of these baselines are missing in the quantitative results.
3. (MS-)SSIM is used as the only evaluation metric throughout the paper. The experiments would be more comprehensive if other metrics such as mean absolute error (MAE) and mean squired error (MSE) are reported.

I would be willing to raise my rating if these issues are properly addressed.

**Questions:**

1. The experiments of equivalence numerical validation in Section 4.1 are a little bit confusing. Could the authors further clarify the specific tasks being evaluated?
2. According to the field experiment, noise is added to the observed data. What about the other scenarios? Did the authors also add noise to the measurements? Such experiments and others with missing traces can further demonstrate the robustness of the proposed method.
3. For the experiments on OpenFWI, some follow-up studies reported better performance (e.g., [1]). The authors may also consider including a few of them as additional baselines.
4. [Minor] The first three paragraphs in Section 3 discussed the advantages of AD and some related works. I think it might be more appropriate to move them to the introduction section or the related work section.
5. [Minor] In Figure 2, there are too many equations with limited context, which may make it difficult for readers to understand the high-level idea.
6. [Minor] The authors mentioned Gaussian filters had been used to generate the initial model. It would be clearer to specify the parameters of the filters rather than using terms such as “heavy” or “slightly”,

[1] Jin, P., Feng, Y., Feng, S., Wang, H., Chen, Y., Consolvo, B., ... & Lin, Y. (2024). An empirical study of large-scale data-driven full waveform inversion. Scientific Reports, 14(1), 20034.

---

> ### Author Response · Authors · 2025-11-20
>
> Thank you! We greatly appreciate these constructive and insightful comments, which improve the quality of the paper, and we have carefully addressed point by point.
>
>
>
> > **W1:** ...  leverage AD such as ... PINN ...
>
> Thank you. It provides an excellent opportunity to clarify the motivation for leveraging lower-level AD instead of prevailing neural networks (NNs).
>
> In short, the novelty of our work does **not** just lie in replacing the adjoint method with AD, but also in **rethinking and addressing the limitations of existing NN‑based methods (e.g., PINNs and neural operators):**
>
> - We carefully analyze the limitations of NN-based methods, particularly their **limited generalizability and interpretability**:
>
> |                          | **NN-based Methods**                                         | **Our AD-based Framework**                                   |
> | :----------------------- | :----------------------------------------------------------- | :----------------------------------------------------------- |
> | **Data Requirements**    | Observed data; Prohibitively large dataset for practical use (e.g., wave propagation in various geology settings, survey geometry settings) | Observed data; A rough initial model (**universally applicable** to any geology and geometry ) |
> | **Detail Recovery**      | Smoothed, potentially missing key anomalies.                 | Good at recovering **high-frequency details.**               |
> | **Physical Consistency** | Enforced via a loss term, but **not** guaranteed in the final result. | **Guaranteed**.                                              |
> | **Interpretability**     | Black-box/Grey-box.                                          | **Completely white-box**. Every step is physically meaningful. |
> | **Generalizability**     | Limited.                                                     | **High** (tested in Nankai).                                 |
>
> - From these issues, we propose a seismic imaging approach that adopts **a low-level AD framework with wave physics**. This approach **embeds seismic wave equations in the forward, white-box propagation**, which is highly **generalizable**, requires **no training data**, and is **capable of capturing high-frequency geological details**.
> - We compare on the **OpenFWI dataset**, showing the best performance on Style-B with the most detailed structures.
> - We successfully apply our method to the Nankai subduction zone, **a challenging real-world case where NN-based methods often fail**.
>
>
>
> Many excellent works based on NNs have emerged, which learn wave propagation from seismic wave simulations using a finite number of geologic models. They also use AD to avoid analytical derivations. **However, when we tried to apply them to real-world imaging tasks from benchmarks, we found that they suffered from challenges due to insufficient number of geologic models in training dataset.** This motivates us to go back to low-level AD with embedded wave equations in this framework.
>
> We identify two primary limitations of NN-based methods:
>
> - **Generalizability**
>
>   - Enormous Dataset Requirements
>
>     Although there are already large public datasets, they are **still far from sufficient** for real‑world problems, mainly for the following reasons:
>
>     - Real subsurface structures are far more complex, varied, and contain sharp discontinuities (faults) than the smooth models in existing training sets. Therefore, one needs a "universal" dataset covering an **infinite number of geological models** in practice.
>
>     - An NN-based model cannot generalize to an unfamiliar geologic structure that is not included in the training dataset.
>
>     - In complex field environments, it’s impossible to set the geometries (source/receiver) the same as the uniform layouts of the training dataset.
>
>
>
>     The **Nankai subduction zone** experiment in our paper is a good case study. **An NN-based approach cannot generalize** here because no training dataset is suitable:
>
>     1. **Structural Mismatch:** The geology of a large-scale subduction zone is structurally different from the smaller models in OpenFWI.
>
>     2. **Scale and Geometry Mismatch:** The scale and layout of the Nankai survey (**131 × 10 km**, 5 km source spacing) are much larger than OpenFWI (0.7 × 0.7 km, 0.14 km source spacing).
>
>
>
>   - Capture Details
>
>     The imaging goal is often to identify **anomalies, which are high-frequency details in the results.** However, the results of **NN‑based methods are often overly smoothed**.
>
> - **Interpretability**
>
>   - NN-based methods are "Black-Box" or "Grey-Box". However, our method  is a **"White-Box"**. Every step is analytically defined and has a physical meaning.
>
>   - Even though physical losses are used to constrain the neural networks, there is no guarantee that the seismic wavefield produced by the network satisfies the PDE at every point.
>
>
>
> Thank you again! We have added and highlighted this comparison in **Introduction**.

---

> > ### Author Response · Authors · 2025-11-20
> >
> > > **W3:** ... mean absolute error (MAE) and mean squired error (MSE) ...
> > >
> > > **Q3**: ... additional baselines.
> > >
> > > [1] Jin, P., Feng, Y., Feng, S., Wang, H., Chen, Y., Consolvo, B., ... & Lin, Y. (2024). An empirical study of large-scale data-driven full waveform inversion. Scientific Reports, 14(1), 20034.
> >
> > Thank you for the insightful comment!
> >
> > In the revised version, we have added additional evaluation metrics, including MAE, MSE and RMSE, to provide a more comprehensive assessment.
> >
> > Regarding **Q3** - the OpenFWI comparison with a great follow-up baseline [1] you recommended, **we reply here as well, along with the new evaluation metrics.**
> >
> >
> >
> > **1. Cross-scenario results** (10 scenarios; Table 1 in the original manuscript; The new results are **highlighted** in **Appendix K. CROSS-SCENARIO RESULTS** ):
> >
> > Results in the table: MAE / MSE / RMSE
> >
> > | **Time Domain**      | **FWI $L_2$**            | **Travel Time**           |
> > | -------------------- | ------------------------ | ------------------------- |
> > | Acoustic             | 0.0401 / 0.0047 / 0.0688 | 0.0701 / 0.0117 / 0.1084  |
> > | SH                   | 0.0642 / 0.0095 / 0.0972 | 0.0645 / 0.0095 / 0.0975  |
> > | P-SV                 | 0.0723 / 0.0117 / 0.1080 | 0.0736 / 0.0148 / 0.1155  |
> > | **Frequency Domain** | **FWI $L_2$**            | **Attenuation**           |
> > | Visco-acoustic       | 0.1034 / 0.0511 / 0.2260 | 0.1154 /  0.0647 / 0.2444 |
> > | Visco-elastic        | 0.0927 / 0.0224 / 0.1495 | 0.0971 / 0.0219 / 0.1481  |
> >
> >
> >
> > **2. OpenFWI results** (with new baseline [1]; revised and **highlighted** in **Table 3,4,5**)
> >
> > After adding the new metrics MAE and RMSE, our AD method shows a significant advantage:
> >
> > | SSIM↑          | BigFWI-B |  BigFWI-M  |  BigFWI-L  | InversionNet | VelocityGAN | UPFWI  |  **Ours (init.)**   |
> > | :------------- | :------: | :--------: | :--------: | :----------: | :---------: | :----: | :-----------------: |
> > | FlatVel-B      |  0.9658  |   0.9729   | **0.9756** |    0.9356    |   0.9556    | 0.8874 |   0.5673 (0.6978)   |
> > | CurveVel-B     |  0.7808  |   0.8053   | **0.8134** |    0.6630    |   0.7111    | 0.6614 |   0.5216 (0.6207)   |
> > | FlatFault-B    |  0.8027  | **0.8137** |   0.8033   |    0.7323    |   0.7552    | 0.6937 |   0.6518 (0.6622)   |
> > | CurveFault-B   |  0.6781  | **0.6896** |   0.6790   |    0.6137    |   0.6033    |   -    |   0.5762 (0.5772)   |
> > | Style-B        |  0.7567  |   0.7600   |   0.7429   |    0.7667    |   0.7249    | 0.6102 | **0.8093** (0.5552) |
> > | Kimberlina-CO₂ |    -     |     -      |     -      |  **0.9872**  |   0.9716    |   -    |   0.9489 (0.7945)   |
> >
> > | MAE↓           | BigFWI-B | BigFWI-M |  BigFWI-L  | InversionNet | VelocityGAN | UPFWI  |  **Ours (init.)**   |
> > | :------------- | :------: | :------: | :--------: | :----------: | :---------: | :----: | :-----------------: |
> > | FlatVel-B      |  0.0233  |  0.0193  | **0.0173** |    0.0304    |   0.0328    | 0.0677 |   0.0395 (0.0402)   |
> > | CurveVel-B     |  0.0933  |  0.0816  | **0.0772** |    0.1448    |   0.1428    | 0.1777 |   0.0813 (0.0958)   |
> > | FlatFault-B    |  0.0710  |  0.0636  |   0.0644   |    0.0965    |   0.0946    | 0.1416 | **0.0544** (0.0689) |
> > | CurveFault-B   |  0.1245  |  0.1161  |   0.1169   |    0.1705    |   0.1583    | 0.3452 | **0.1034** (0.1313) |
> > | Style-B        |  0.0553  |  0.0538  |   0.0563   |    0.0557    |   0.0649    | 0.1702 | **0.0399** (0.0757) |
> > | Kimberlina-CO₂ |    -     |    -     |     -      |  **0.0061**  |   0.0119    |   -    |   0.0103 (0.0193)   |
> >
> > | RMSE↓          | BigFWI-B |  BigFWI-M  |  BigFWI-L  | InversionNet | VelocityGAN | UPFWI  |  **Ours (init.)**   |
> > | :------------- | :------: | :--------: | :--------: | :----------: | :---------: | :----: | :-----------------: |
> > | FlatVel-B      |  0.0696  |   0.0621   | **0.0584** |    0.0680    |   0.0787    | 0.1493 |   0.0718 (0.0879)   |
> > | CurveVel-B     |  0.2154  |   0.2006   | **0.1947** |    0.3111    |   0.2611    | 0.3179 |   0.2073 (0.2251)   |
> > | FlatFault-B    |  0.1321  | **0.1259** |   0.1269   |    0.1636    |   0.1553    | 0.2220 |   0.1283 (0.1398)   |
> > | CurveFault-B   |  0.2027  |   0.1954   |   0.1960   |    0.2507    |   0.2336    | 0.5010 | **0.1918** (0.2050) |
> > | Style-B        |  0.0876  |   0.0867   |   0.0908   |    0.0860    |   0.0979    | 0.2609 | **0.0406** (0.0921) |
> > | Kimberlina-CO₂ |    -     |     -      |     -      |    0.0374    |   0.0387    |   -    | **0.0195** (0.0403) |

---

> ### Author Response · Authors · 2025-11-20
>
> > **W2:** In the experiment section, the author stated that there are two baselines (ADFWI and a Matlab-based visco-acoustic wave equation solver). However, the performance of these baselines are missing in the quantitative results.
>
> We appreciate your careful reading and apologize for the confusion caused by our wording.
>
> We would like to clarify that the term “***baselines***” in our experimental section was not intended to refer to baseline models for performance comparison in the deep-learning sense. Instead, it denotes the **baseline implementations (or basic codes)** upon which our proposed AD framework is built. Their code **has different functions from ours**.
>
> Therefore, there are no quantitative comparisons between these baseline implementations and our method.
>
> The extension from Matlab-based visco‑acoustic wave equation solver  (Amini & Javaherian, 2011):
>
> |           | Matlab-based visco‑acoustic wave equation solver | Ours                                                         |
> | --------- | ------------------------------------------------ | ------------------------------------------------------------ |
> | Language  | Matlab (not supporting AD)                       | **PyTorch (supporting AD and directly integrating with deep learning methods)** |
> | Wave Type | visco‑acoustic                                   | Visco‑acoustic, **Visco‑elastic**                            |
> | Function  | Forward (only for modeling)                      | Forward, **Inversion**                                       |
>
> The extension from ADFWI (Liu et al., 2024):
>
> |           | ADFWI          | Ours                                                         |
> | --------- | -------------- | ------------------------------------------------------------ |
> | Wave Type | Acoustic, P–SV | Acoustic, P–SV, **SH (love wave), Visco‑acoustic, Visco‑elastic** |
> | Domain    | Time           | Time, **Frequency**                                          |
> | Misfit    | $L_2$ waveform | **Arbitrary, e.g., waveform, travel time, and attenuation...** |
>
> Thanks again for the your insightful comment. In the revised manuscript, we have replaced the ambiguous term *“**baseline**”* with *“**baseline codes**”* to avoid confusion and improve clarity.

---

> ### Author Response · Authors · 2025-11-20
>
> Following replies of W3 and Q3:
>
> **3. New experiments: robustness of all metrics to high‑frequency noise**
>
> The limitation of our method in SSIM comparison has already been mentioned in the manuscript: while our method captures fine details, it also introduces high‑frequency noise, to which SSIM is highly sensitive.
>
> **To compare the robustness of all metrics (SSIM, MS‑SSIM, MAE, and RMSE) to high‑frequency noise, we conducted new experiments.**
>
> |                          | Gentle Blur | Strong Blur |
> | ------------------------ | ----------- | ----------- |
> | High‑frequency Anomalies | More        | Less        |
> | High‑frequency Noise     | More        | Less        |
>
> The experimental results show that SSIM tends to select the model with the strongest blur — the one where **both high‑frequency noise and high‑frequency anomalies are** **removed**. **That is, when a method can detect anomalies but also introduces some high‑frequency noise (such as our method), SSIM tends to produce a low value, thereby misjudging the method.** This metric may be non‑robust and impractical, as seismic imaging without anomaly details fails.
>
> In contrast, the other metrics tend to select models that preserve high‑frequency anomalies, demonstrating greater robustness to high‑frequency noise.
>
> The new experiment is added and highlighted in **Appendix F. SSIM’S VULNERABILITY TO HIGH-FREQUENCY ARTIFACTS**.
>
>
>
>
>
> Thank you very much for your suggestion to include additional metrics, which has helped us to demonstrate the effectiveness of our method more clearly! We also sincerely appreciate your introduction of the novel baseline [1] for comparison!
>
>
>
>
>
> > **Q1**: The experiments of equivalence numerical validation in Section 4.1 are a little bit confusing. Could the authors further clarify the specific tasks being evaluated?
>
> Thank you for your comments. We appreciate your observation that the explanation in Numerical Validation was not sufficiently clear.
>
> - Purpose: to numerically verify the **numerical equivalence between adjoint method and our AD method**, after our theoretical proof.
> - Models: anomaly synthetic models, the Marmousi2 model, and the OpenFWI-B family
> - Wave type and domain: acoustic wave in the time domain and Love wave in the frequency domain
> - Step:
>   - For a given model and one forward simulation, we **compute the gradient of misfit with respect to the model parameters using two methods**: (1) a manually implemented traditional adjoint‑state solver; (2) our AD framework.
>   - Then we compare the two resulting gradients by computing their **difference (using both norm and max of the difference)**, **correlation**, and **SSIM**, to verify their **numerical equivalence**.
> - Result: across all tested scenarios, **all metrics indicate numerical equivalence**.
>   - Correlations and SSIM values are very close to 1 (difference $< 10^{-4}$)
>   - Difference Norm and Difference Max consistently remain on the order of  $10^{-10}$,  which almost reaches floating‑point precision.
>
> We have revised and highlighted the above procedure in **Appendix C. EQUIVALENCE NUMERICAL VALIDATION.**

---

> ### Author Response · Authors · 2025-11-20
>
> > **Q2**: According to the field experiment, noise is added to the observed data. What about the other scenarios? Did the authors also add noise to the measurements? Such experiments and others with missing traces can further demonstrate the robustness of the proposed method.
>
> Thank you for your insightful comment.
>
> For experiments except OpenFWI, we added random noise to the observed data. To better simulate a real‑world task, we used a **high noise level of 1% maximum amplitude with a low SNR.** For example, in acoustic‑wave experiments, the SNR drops rapidly as the noise level increases; at a 1% noise level, it is only **9.21 dB**:
>
> | Noise level (% of max. amplitude) | 0%   | 0.1%  | 0.3%  | 0.5%  | 0.7%  | 1%                                |
> | --------------------------------- | ---- | ----- | ----- | ----- | ----- | --------------------------------- |
> | SNR (dB)                          | -    | 29.21 | 19.67 | 15.23 | 12.32 | **9.21(used in our experiments)** |
>
> Even under high noise levels, our method can still image the model details, demonstrating its robustness to noise.
>
> As for the OpenFWI benchmark test, the original paper did not mention any noise in the data; therefore, we followed the same setting for fair comparison with the benchmark.
>
>
>
> We fully agree that additional experiments with **noise and missing traces** are valuable for testing robustness. We have conducted new experiments using the acoustic wave on the OpenFWI Style-B, and the results are shown below:
>
> | Noise level | 0%     | 0.1%   | 0.3%   | 0.5%   | 0.7%   | 1%     |
> | ----------- | ------ | ------ | ------ | ------ | ------ | ------ |
> | SNR(dB)     | -      | 30.27  | 20.71  | 16.29  | 13.37  | 10.27  |
> | SSIM        | 0.7820 | 0.7322 | 0.6567 | 0.6086 | 0.6000 | 0.5966 |
> | MAE         | 0.0314 | 0.0358 | 0.0428 | 0.0458 | 0.0474 | 0.0460 |
> | RMSE        | 0.0443 | 0.0509 | 0.0571 | 0.0609 | 0.0630 | 0.0614 |
>
> | Missing traces (%) | 0%     | 1%     | 4%     | 10%    | 20%    | 50%    | 70%    | 90%    |
> | ------------------ | ------ | ------ | ------ | ------ | ------ | ------ | ------ | ------ |
> | SSIM               | 0.7820 | 0.7793 | 0.7808 | 0.7779 | 0.7806 | 0.7754 | 0.7551 | 0.6692 |
> | MAE                | 0.0314 | 0.0315 | 0.0314 | 0.0316 | 0.0317 | 0.0324 | 0.0330 | 0.0421 |
> | RMSE               | 0.0443 | 0.0448 | 0.0445 | 0.0446 | 0.0441 | 0.0453 | 0.0461 | 0.0562 |
>
>
>
> These new experiments are added and highlighted in **Appendix L. ROBUSTNESS TEST.**
>
> Thank you for your suggestions!
>
>
>
>
>
> > **Q4**: [Minor] The first three paragraphs in Section 3 discussed the advantages of AD and some related works. I think it might be more appropriate to move them to the introduction section or the related work section.
>
> Thank you. We have added a Related Work section.
>
>
>
> > **Q5**: [Minor] In Figure 2, there are too many equations with limited context, which may make it difficult for readers to understand the high-level idea.
>
> Thank you for your careful observation and insightful suggestion!
>
>  We would like to respectfully clarify that the use of many equations in Figure 2 is intentional. Our aim is not for readers to follow every equation in detail, but rather to help them immediately grasp how complex and challenging the analytical derivation of the adjoint method is at the first glance.
>
>
>
> > **Q6**: [Minor] The authors mentioned Gaussian filters had been used to generate the initial model. It would be clearer to specify the parameters of the filters rather than using terms such as “heavy” or “slightly”,
>
> Thank you. The parameters are shown below:
>
> |                    | Marmousi2 | Q anomaly | OpenFWI-B | Kimberlina-CO2 |
> | ------------------ | --------- | --------- | --------- | -------------- |
> | Gaussian Parameter | 3         | 10        | 5.5       | 7              |
>
> We have added them in the revised manuscript.

---

### Author Response · Authors · 2025-12-03
**Global Response**

We sincerely appreciate the reviewers’ insightful comments and the hard work of the AC, SAC, and PC. We have carefully replied to all concerns point by point and revised the manuscript accordingly. A summary of the major updates is below:

1. **Motivation and Novelty (Reviewer 7KA1, 4MRU)**

   - Motivation

     Many excellent works based on neural networks have emerged, showing great performance on benchmarks (e.g., OpenFWI). However, when applied to **real‑world tasks**, we found that their **limited practicality**, particularly due to **(1) limited generalizability** (requirement for a huge training dataset; fail to generalize to the Nankai subduction zone in our paper) and **(2) weak interpretability** (black / grey box).

   - Novelty

     To address this issue, we propose **a white-box alternative to black-box neural networks (NN) by switching to physics-embedded Recurrent Neural Network (RNN) with Automatic Differentiation (AD)**, which has previously been used to avoid analytical gradient derivations in full wave inversion (FWI). Building on this insight, **our method extends to time and frequency domains with various misfit (loss) functions, thus removing the need for large training datasets**. Moreover, as a physics‑based approach, our framework provides **better structural detail recovery and stronger geophysical consistency**.

     Specifically,

     - We explicitly formulate every step of our white‑box RNN seismic imaging. We **proved theoretically and numerically for the first time** that the gradients from AD of RNN are equivalent to those from the commonly used analytical adjoint method, **regardless of the domain, wave equation, or choice of misfit function**.
     - We validate our new framework through experiments across **ten diverse scenarios,** OpenFWI benchmark tests (where it **outperforms NN‑based methods on detail‑rich models**), and a **field test** in the Nankai subduction zone.
     - We developed a customizable PyTorch-based AD seismic imaging framework, the first to support both time and frequency domains. This design allows it to be directly used or embedded into deep learning workflows.



     Our work shows that AD with RNN is not just a tool to avoid manual gradient derivation in traditional adjoint methods, but also provides **unifying capability, strong generalizability, and high interpretability in inverse problems, suggesting broader applications in AI for science.**





2. **Additional SOTA Baselines and Metrics (Reviewer 13jS)**

   - We added MAE and RMSE in addition to SSIM for all experiments.
   - Following Reviewer 13jS’s suggestion, we added the **recent NN-based approach BigFWI** (Jin et al., 2024), besides original InversionNet, VelocityGAN and UPFWI.

   The results (Tables 3, 4, 5, and Appendix K) show that our method performs better than NN‑based methods on complex models with rich structural details (e.g., Style‑B), whereas NN‑based methods perform better on uniform models (e.g., FlatVel‑B):

| ✓ marks ours the best | SSIM | MAE  | RMSE |
| --------------------- | :--: | :--: | :--: |
| FlatVel‑B             |      |      |      |
| CurveVel‑B            |      |      |      |
| FlatFault‑B           |      |  ✓   |      |
| CurveFault‑B          |      |  ✓   |  ✓   |
| Style‑B               |  ✓   |  ✓   |  ✓   |
| Kimberlina‑CO₂        |      |      |  ✓   |



3. **Theoretical Equivalence Between AD and the Adjoint Method Is Well‑Known? (Reviewer 7KA1, 4MRU)**

   We clarified that the equivalence in our paper is ***not*** trivial and **goes beyond prior proofs** for three reasons:

   - handling **generalized misfit functionals** (previous work was limited to the L2 norm)
   - providing time‑domain derivations in the **step‑by-step formulation** (previous work was largely limited to pure **matrix‑form** AD)
   - introducing **Wirtinger calculus** to handle previously challenging **complex‑valued frequency‑domain** problems (where prior work was limited to real‑valued time‑domain formulations)



4. **Robustness Tests (Reviewer 13jS)**

   Even with **70% missing traces or SNR dropping to ~20 dB**, our method can still successfully image the basic structures. Added in **Appendix L**.

   | **Noise (%)** | **0%** | **0.1%**  | **0.3%**  | **1%**    |
   | ------------- | ------ | --------- | --------- | --------- |
   | **SNR(dB)**   | -      | **30.27** | **20.71** | **10.27** |
   | SSIM          | 0.7820 | 0.7322    | 0.6567    | 0.5966    |
   | MAE           | 0.0314 | 0.0358    | 0.0428    | 0.0460    |
   | RMSE          | 0.0443 | 0.0509    | 0.0571    | 0.0614    |

   | Missing traces (%) | 0%     | 20%    | 50%    | 70%    | 90%    |
   | ------------------ | ------ | ------ | ------ | ------ | ------ |
   | SSIM               | 0.7820 | 0.7806 | 0.7754 | 0.7551 | 0.6692 |
   | MAE                | 0.0314 | 0.0317 | 0.0324 | 0.0330 | 0.0421 |
   | RMSE               | 0.0443 | 0.0441 | 0.0453 | 0.0461 | 0.0562 |

---

### Meta-Review · Area_Chair_cFzg · 2026-01-06

**Summary:**

This paper is about a specialized domain, Seismic Tomography. The authors use an AD approach to derive a simplified and unified set of equations for seismic tomography that avoids the manual, derivative calculations common in traditional adjoint-state methods. The authors present a theoretical equivalence proof between AD-computed gradients and adjoint gradients, generalizing this demonstration to the frequency domain and different misfit functions by means of Wirtinger calculus. While the paper demonstrates some practical meanings within seismic tomography, its narrow application domain might limit its appeal to the broader ICLR community.

**Reviewer Concerns:**

The main criticisms about rejection revolve around the limited methodological novelty compared to machine learning. Comments Reviewers 4MRU and 7KA1 suggest that AG replacing adjoint derivation by hand with manual AD itself is more of an engineering choice than a fundamental algorithmic advancement. reviewers argued that this is still a very physics based full waveform inversion without much of learning. Accordingly, the consensus is that it is more appropriate for computational geophysics venues (reviewer 4MRU, 7KA1).

**Reviewer Scores:**

Reviewer 4MRU: unconvinced of the novelty, would remain 2

Reviewer 7KA1: kept concerns regarding the incremental contributions,   2

Reviewer 13jS: fundamental scope issues remain, and would give 4

Reviewer YP9p: is highly positive on theoretical proofs and gives 8, but with a fair confidence of 3

---

### Decision · Program_Chairs · 2026-01-26

Reject